# MoS: Unleashing Parameter Efficiency of Low-Rank Adaptation with Mixture of Shards

**Sheng Wang**[*], **Liheng Chen**[*], **Pengan Chen**
School of Computing and Data Science
The University of Hong Kong
{u3009618, clh648, cpa2001}@connect.hku.hk

**Jingwei Dong**
School of Electrical Engineering
Xi'an Jiaotong University
dongjingwei@stu.xjtu.edu.cn

**Boyang Xue, Jiyue Jiang**
Department of Systems Engineering and Engineering Management
The Chinese University of Hong Kong
byxue@se.cuhk.edu.hk, jiangjy@link.cuhk.edu.hk

**Lingpeng Kong, Chuan Wu**
School of Computing and Data Science
The University of Hong Kong
{lpk, cwu}@cs.hku.hk

## Abstract

The rapid scaling of large language models necessitates more lightweight fine-tuning methods to reduce the explosive GPU memory overhead when numerous customized models are served simultaneously. Targeting more parameter-efficient low-rank adaptation (LoRA), parameter sharing presents a promising solution. Empirically, our research into high-level sharing principles highlights the indispensable role of differentiation in reversing the detrimental effects of pure sharing. Guided by this finding, we propose Mixture of Shards (MoS), incorporating both inter-layer and intra-layer sharing schemes, and integrating four nearly cost-free differentiation strategies, namely subset selection, pair dissociation, vector sharding, and shard privatization. Briefly, it selects a designated number of shards from global pools with a Mixture-of-Experts (MoE)-like routing mechanism before sequentially concatenating them to low-rank matrices. Hence, it retains all the advantages of LoRA while offering enhanced parameter efficiency, and effectively circumvents the drawbacks of peer parameter-sharing methods. Our empirical experiments demonstrate approximately $8\times$ parameter savings in a standard LoRA setting. The ablation study confirms the significance of each component. Our insights into parameter sharing and MoS method may illuminate future developments of more parameter-efficient finetuning methods. The code is officially available at https://github.com/Forence1999/MoS.

## 1 Introduction

With the remarkable capabilities, large language models (LLMs) (Dubey et al., 2024; Team et al., 2023; Jiang et al., 2023) are adapted for various downstream applications (Shao et al., 2023; 2024). Among parameter-efficient finetuning (PEFT) methods (Houlsby et al., 2019; Liu et al., 2022), low-rank adaptation (LoRA) (Hu et al., 2021) reparameterizes the weight updates with two trainable low-rank matrices, and has gained the most popularity with reduced resource consumption, competitive performance, low-cost task switching, etc. Nevertheless, with the continuous applicability of scaling law (Achiam et al., 2023), the size of LLMs increases rapidly for even better performance, intensifying the need for more lightweight finetuning methods. For instance, GPT4o allows customization to its vast user base[1], incurring intense challenges to serve numerous customized models

---

[*]Equal Contribution.
[1]https://openai.com/index/gpt-4o-fine-tuning/

simultaneously. Assuming a scenario with a Llama2-70B-sized model and 10,000 active users, each allocated a LoRA module with the rank of 16, only the parameters of LoRAs would occupy 3.36 TB of GPU memory, an onerous burden for service providers. This underscores the need for a more lightweight solution than LoRA, while retaining its advantages.

As a promising solution, parameter sharing has been extensively explored (Kopiczko et al., 2023; Renduchintala et al., 2023; Wang et al., 2024b). Specifically, VeRA (Kopiczko et al., 2023) adopts inter-layer sharing of frozen matrices with trainable vectors; however, it results in limited model capacity and necessitates an excessively high rank. Subsequently, Tied LoRA (Renduchintala et al., 2023) mitigates these constraints by permitting shared matrices to be trainable. Yet, its tying mechanism restricts applicability to linear layers of different dimensions. More recently, PRoLoRA (Wang et al., 2024b) showcases enhanced parameter efficiency by reusing submatrices within a single linear layer, thereby circumventing the above restrictions. However, its limited focus on intra-layer sharing constrains efficiency potential, and promises a hybrid approach integrating both intra-layer and inter-layer sharing. Besides, while these methods, with their intuitive designs, have shown the benefits of parameter sharing on enhanced parameter efficiency, the fundamental principles have not been discussed yet to guide the future development of parameter sharing strategies.

Targeting enhanced parameter efficiency, we first analyze the factors influencing parameter sharing performance in the context of LoRA. Our empirical findings indicate that improving the parameter efficiency of LoRA requires a careful balance between sharing and differentiation. While parameter sharing can significantly decrease the number of trainable parameters, the negative effects (*i.e.*, possible performance degradation) of excessive sharing necessitate differentiation strategies to mitigate these drawbacks. Guided by this insight, we propose a more parameter-efficient LoRA-based method called Mixture of Shards (MOS). It features a global sharing scheme to achieve a higher sharing ratio, along with four nearly cost-free differentiation techniques: subset selection, pair dissociation, vector sharding, and shard privatization. Briefly, our method selects a designated number of shards from global pools with a Mixture-of-Experts (MoE)-like routing mechanism (Jacobs et al., 1991), and sequentially concatenate them to construct the low-rank matrices.

Empirically, we validate its superior performance compared to peer methods in scenarios with fixed trainable parameter count, and demonstrate an eightfold parameter reduction in performance-targeted situations. A thorough ablation analysis of each differentiation strategy reveals that all contribute to unleashing the parameter efficiency of MOS, despite their nearly cost-free property. Notably, pair dissociation and shard privatization boost the efficiency more significantly through increased combination diversity and exclusive differentiation, respectively, whereas vector sharding provides only limited gains in diversity. With the intuitive design, our method retains all the advantages of LoRA while circumventing the drawbacks of peer methods. The markedly improved parameter efficiency alleviates the burden on service providers when numerous LoRA-based customized models are served simultaneously.

In summary, our main contributions are as follows:

- We introduce a more parameter-efficient finetuning method named Mixture of Shards (MOS), which facilitates the concurrent serving of numerous customized models with the saliently reduced GPU memory overhead.
- Our exploration into high-level sharing principles highlights the balance between sharing and differentiation to enhance parameter efficiency, providing valuable insights for the future advancement of parameter-sharing methods.
- Extensive experiments empirically confirm the significant parameter savings achieved by MOS, and validate the necessity and relative importance of each differentiation strategy.
- To the best of our knowledge, we are the first to apply MoE-like mechanism for parameter savings in a single-task LoRA. Besides, our dissociated perspective may also inspire further parameter reduction for LoRA-based multitasking scenarios.

## 2 SHARING & DIFFERENTIATION

Drawing inspiration from the inter-layer shared VeRA (Kopiczko et al., 2023) and intra-layer shared PRoLoRA (Wang et al., 2024b), we seek to create a unified sharing mechanism that leverages both

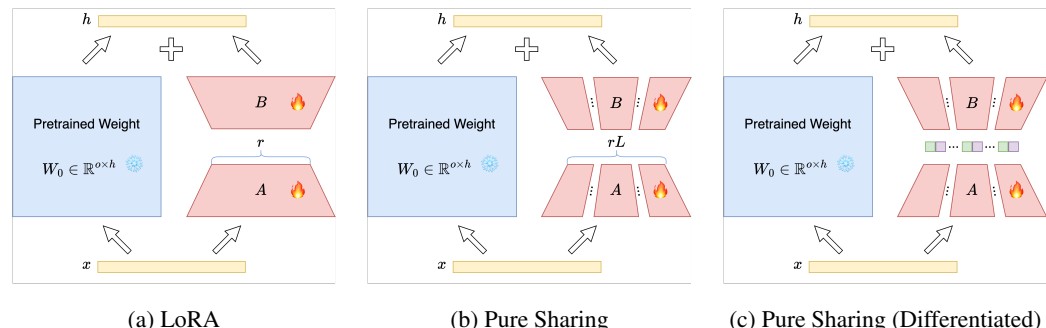

(a) LoRA              (b) Pure Sharing          (c) Pure Sharing (Differentiated)

Figure 1: Illustration of vanilla LoRA, pure sharing, and differentiated pure sharing methods. The rank of vanilla LoRA is denoted as $r$, while that of pure sharing is boosted to $rL$ through the aggregation of trainable parameters among $L$ Transformer blocks. In Figure 1c, the colored squares between the low-rank matrices $\mathbf{A}$ and $\mathbf{B}$ signify scalars and boolean values for random scaling and subset selection strategies, respectively.

perspectives, promising further enhanced parameter efficiency. To guide this development more clearly, we start with reparameterizing the vanilla LoRA, followed by analyzing the impact of high-level factors (*i.e.*, sharing and differentiation) on model performance under a specified number of trainable parameters. All relevant approaches are illustrated in Figure 1 for an intuitive comparison.

**Reparameterization of LoRA.** LoRA (Hu et al., 2021) efficiently adapts LLMs by updating the frozen pretrained weights with low-rank decomposition. Specifically, given a pretrained weight $\mathbf{W}_0 \in \mathbb{R}^{o \times h}$, its weight update $\Delta \mathbf{W}$ is approximated as the product of two trainable matrices $\mathbf{A} \in \mathbb{R}^{r \times h}$ and $\mathbf{B} \in \mathbb{R}^{o \times r}$, where the rank $r$ is times smaller than $h$ and $o$ for parameter saving. Similar to Kopiczko et al. (2023), this product can be further reparameterized as the sum of outer products of the corresponding row and column vectors. Formally, the whole process can be presented as:

$$\mathbf{W} = \mathbf{W}_0 + \Delta \mathbf{W} = \mathbf{W}_0 + \mathbf{B}\mathbf{A} = \mathbf{W}_0 + \sum_{i=1}^{r} \mathbf{b}_i \otimes \mathbf{a}_i, \tag{1}$$

where $\mathbf{W}$ is the updated weight, and $\mathbf{a}_i$ and $\mathbf{b}_i$ denote the $i$-th row and column vectors of low-rank matrices $\mathbf{A}$ and $\mathbf{B}$, respectively. This reparameterization trick enables the weight update to be the linear combination of outer products of multiple vector pairs, which will be extensively utilized in our subsequent analysis. Generally, LoRA will be applied to each linear layers in all the Transformer blocks of a model (Vaswani et al., 2017) for better performance (Dettmers et al., 2023), which is also applied to our analysis by default.

**Pure parameter sharing does not necessarily boost up the parameter efficiency of LoRA.** Although VeRA and PRoLoRA achieve better performance than LoRA via parameter sharing, we demonstrate that this causality is not universally applicable. To facilitate our subsequent comparisons, we introduce a straightforward sharing scheme, detailed in Sec. 3.1. Briefly, this scheme shares the low-rank matrices $\mathbf{A}$ and $\mathbf{B}$ across all Transformer blocks for each type of linear layer. As shown in Table 1, we set the same low-level trainable parameter budget for all methods, eliminating the disturbance of parameter redundancy and enabling clear comparison of their representation

Table 1: Results of LLaMA2-7B with different sharing and differentiation methods across diverse instruction following datasets. "+ Random Scaling" and "+ Subset Selection" denote the individual integration of them into the "Pure Sharing" scheme, respectively. The details of these benchmarks are elaborated in Sec. 4.

| Method | Rank | # Param. | MMLU | BBH | GSM8K | TyDi QA | | HumanEval | Avg. |
|---|---|---|---|---|---|---|---|---|---|
| | | | EM | EM | EM | F1 | EM | P@1 | |
| LoRA | 2 | 5.00M | 44.77 | 36.22 | 26.28 | 48.67 | 35.70 | 18.24 | 34.98 |
| Pure Sharing | 64 | 5.00M | 43.61 | 35.15 | 26.54 | 49.12 | 36.04 | 15.53 | 34.33 |
| + Random Scaling | 64 | 5.00M | 43.98 | 35.30 | 29.04 | 49.03 | 35.73 | 15.58 | 34.77 |
| + Subset Selection | 64 | 5.00M | 45.56 | 36.76 | 28.18 | 50.33 | 37.22 | 18.64 | 36.12 |

capabilities. Despite several times higher rank, which is suggested for better performance (Wang et al., 2024b), this pure sharing scheme outperforms LoRA on GSM8K and Tydi QA tasks, but underperforms on others. This variability indicates that excessive sharing may hinder model performance, negating the positive effects of increased rank. In other words, sharing mechanism is not the only crucial element of existing methods for enhanced parameter efficiency.

**Differentiation reverses the detrimental effects of pure sharing mechanism.** Inspired by the varying rank demands in different positions (Zhang et al., 2023), we hypothesize that the pure sharing scheme, reusing the same matrices among all Transformer blocks, contradicts different representation capacity across blocks, potentially deteriorating the performance. To verify this, we devise the following two schemes to differentiate the shared matrices:

1. **Random Scaling.** Random scaling adds scalars to each vector pair, which can be formulated as $\mathbf{BA} = \sum_{i=1}^{r} s_i \cdot \mathbf{b}_i \otimes \mathbf{a}_i$, where $s_i$ is sampled from a normal distribution and frozen after initialization. Hence, this operation does not introduce extra trainable parameters, but different blocks will optimize the vectors associated with their own larger scalars despite the same shared whole matrices, since these vectors influence the representation capacity more significantly than others.

2. **Subset Selection.** Instead of scaling the vector pairs, this scheme selects a specific number of vector pairs from the shared matrices. Specifically, the scalars in the random scaling method are substituted with Boolean values. Vectors corresponding to those marked as true are selected, while the others are neglected. This ensures that, in most cases, any two blocks only share a subset of vector pairs so that their representations are differentiated by other distinct pairs.

As shown in Table 1, both differentiation strategies further improve model performance over the pure sharing scheme. Specifically, random scaling can enhance the performance of pure sharing in nearly all tasks, and ultimately achieves slightly better average results. However, it still underperforms LoRA apparently. In contrast, subset selection, adopting more aggressive differential operations, reverses the degradation and consistently surpasses both pure sharing and LoRA across all benchmarks. On average, its performance exceeds that of LoRA over one percent, highlighting the importance of differentiation. Further experiments and analysis on LLaMA3.2-3B model (Dubey et al., 2024) are provided in Appendix B.2.

In summary, enhancing the parameter efficiency of LoRA involves a delicate balance between sharing and differentiation. On one hand, parameter sharing can significantly reduce the number of trainable parameters; on the other hand, the detrimental impacts of sharing necessitate differentiation schemes to counteract them. This insight will inform our subsequent design efforts.

## 3 METHOD

Based on the above analysis, we follow the guide of sharing and differentiation, and propose a more lightweight LoRA-based method, named Mixture of Shards (MoS). Briefly, it selects a specific number of shards from global pools with a MoE-like routing mechanism, and combine them sequentially to form the low-rank matrices of LoRA. It features a global sharing scheme for a higher sharing ratio, and four nearly cost-free differentiation measures, namely subset selection, pair dissociation, vector sharding, and shard privatization. Their details will be elaborated below, followed by the advantage analysis.

### 3.1 GLOBAL SHARING

To maximize the parameter efficiency, we introduce a global sharing mechanism, incorporating VeRA-like inter-layer sharing and PRoLoRA-like intra-layer sharing methods. As illustrated in Figure 2a, this mechanism centers on a global pool for each type of linear layer. Initially, each pool consists of multiple independently initialized and trained vector pairs, while each vector pair within the low-rank matrix pairs (*i.e.*, $\mathbf{A}$ and $\mathbf{B}$) for each layer is sampled from this pool. The "pure sharing" in Sec. 2 can be formed when all vector pairs from the pool are selected at each block. In this case, the weight update can be formulated as

$$\Delta \mathbf{W} = \mathbf{BA} = \mathbf{B}^p \mathbf{A}^p, \tag{2}$$

where $\mathbf{A}^p \in \mathbb{R}^{eL \times h}$ and $\mathbf{B}^p \in \mathbb{R}^{o \times eL}$ represents the globally shared low-rank matrices for one linear layer type[2], while $L$ and $e$ denote the number of Transformer blocks and an equivalent rank to that of LoRA in terms of trainable parameter count, respectively. Due to the extremely high sharing ratio (*i.e.*, $L$), a small number of parameters can yield a remarkably high rank, hinting positive effects on performance within a certain range (Wang et al., 2024a;b). Taking the LLaMA2-7B model with 32 blocks (Touvron et al., 2023) as an example, pure sharing can raise the rank from 2 to 64 with the same trainable parameter count as LoRA, indicating its great potential for parameter reduction.

## 3.2 SUBSET SELECTION

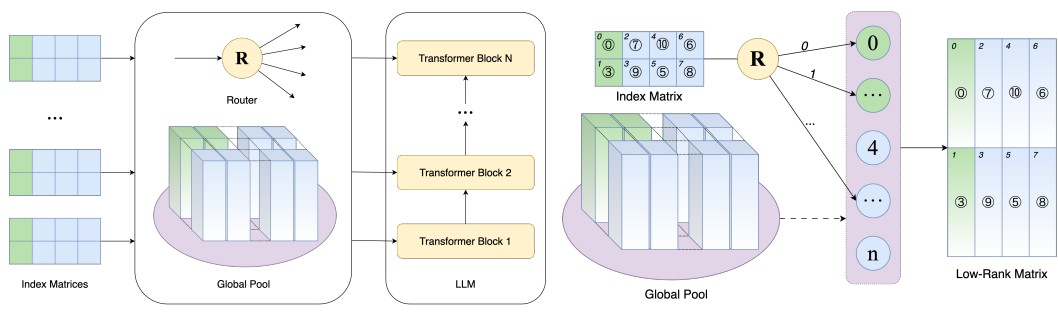

(a) Inter-Layer Sharing  (b) Intra-Layer Sharing

Figure 2: Illustration of MOS from the perspectives of inter-layer and intra-layer sharing. (a) Across layers, each layer retrieves shards from the same global pool, utilizing an independent index matrix and a MoE-like router "R". (b) Within each layer, the shard retrieval process is visualized in details with the number of shards per vector $l$, rank $r$, pool size as 2, 4, and n, respectively. The circled numbers denote the shard indices in the global pool, while the small italicized numbers indicate the shard positions within the low-rank matrix. Blue highlights shared components, whereas green indicates privatized ones.

As discussed in Sec. 2, while pure sharing scheme may lead to performance decline due to the shared weight updates across Transformer blocks, differentiation plays an indispensable role in reversing the detrimental effects. Compared to random scaling, subset selection demonstrates a more pronounced performance improvement, and is thus incorporated into our design for differentiation. Let $r$ denote the number of vector pairs chosen from the shared pools, signifying the rank of each low-rank matrix during both training and inference. The weight update for the $k$-th block can be expressed as follows:

$$\Delta \mathbf{W}^k = \mathbf{B}^k \mathbf{A}^k = \mathbf{B}^p \Lambda^k \mathbf{A}^p = \sum_{i=1}^{eL} m_i^k \cdot \mathbf{b}_i^p \otimes \mathbf{a}_i^p, \tag{3}$$

where the boolean value $m_i^k$ represents the $i$-th diagonal element of the diagonal matrix $\Lambda^k$ for the $k$-th block. This value is randomly sampled during initialization, remains fixed during the finetuning process, and satisfies the condition $\sum_{i=1}^{eL} m_i^k = r$. Additionally, $\mathbf{a}_i^p$ and $\mathbf{b}_i^p$ denote the $i$-th row and column vectors of $\mathbf{A}^p$ and $\mathbf{B}^p$, respectively.

## 3.3 PAIR DISSOCIATION

In prior research, such as Valipour et al. (2022); Kopiczko et al. (2023), vector pairs have been regarded as the fundamental units of LoRA by default. To foster greater differentiation, we propose a vector pair disassociation strategy that is straightforward and cost-effective, requiring no additional trainable parameters while yielding significant improvements in our empirical analysis. Specifically, instead of a single shared pool for vector pairs (*i.e.*, pairs of $\mathbf{a}_i^p$ and $\mathbf{b}_i^p$), we take decoupled vectors as the basic units, and separate $\mathbf{a}_i^p$ and $\mathbf{b}_i^p$ into two distinct pools. Hence, within the $k$-th block, the vectors that form matrices $\mathbf{A}^k$ and $\mathbf{B}^k$ will be sampled independently. Moreover, this separation naturally enables two ordered index vectors, $\mathbf{I}_a^k \in \mathbb{N}^r$ and $\mathbf{I}_b^k \in \mathbb{N}^r$, instead of unordered boolean

---

[2]Due to the identical operations, the notation for different types of linear layers is omitted for clarity.

vectors, to access vectors from the pools. Both the dissociation and the ordered indices significantly enhance the combinatorial diversity of $\mathbf{A}^k$ and $\mathbf{B}^k$. Formally, this process can be formulated as

$$\Delta\mathbf{W}^k = \mathbf{B}^k\mathbf{A}^k = \text{Route}^c(\mathbf{B}^p, \mathbf{I}_b^k)\,\text{Route}^r(\mathbf{A}^p, \mathbf{I}_a^k), \tag{4}$$

where $\text{Route}^c(\cdot)$ and $\text{Route}^r(\cdot)$ denote the retrieval of column and row vectors, respectively, based on the given indices.

## 3.4 Vector Sharding

Drawing inspiration from database sharding[3], we further refine the basic unit of the shared pool to vector shards, where $l$ sampled shards are concatenated into a single vector. As illustrated in Figure 2b, two vectors corresponding to the indices "0" and "3" are retrieved and combined to form the first rank of the low-rank matrix. Similar to vector disassociation, this nearly cost-free operation further promotes the diversity and differentiation of low-rank matrices. Subsequent experiments will confirm its beneficial effects. The mathematical formulation remains unchanged, with the exception that the dimensions of $\mathbf{I}_a^k$ and $\mathbf{I}_b^k$ are adjusted to $l \times r$.

## 3.5 Shard Privatization

As another differentiation strategy, we partition the global pool into two segments: public and private. As indicated by their names, the public segment remains shared, whereas the private part is exclusively accessible to only one matrix. This privatization improves the differentiation of weight updates among blocks, as expressed by

$$\Delta\mathbf{W}^k = \mathbf{B}^k\mathbf{A}^k = \text{Route}^c(\text{Concat}(\mathbf{A}^{pub}, \mathbf{A}^{pri}), \mathbf{I}_b^k)\,\text{Route}^r(\text{Concat}(\mathbf{B}^{pub}, \mathbf{B}^{pri}), \mathbf{I}_a^k), \tag{5}$$

where $\mathbf{A}^p$ in Eq. 4 is substituted by the concatenation $\text{Concat}(\cdot)$ of the public pool $\mathbf{A}^{pub}$ and the private pool $\mathbf{A}^{pri}$. Similarly, $\mathbf{B}^p$ is replaced by the he concatenation of $\mathbf{B}^{pub}$ and $\mathbf{B}^{pri}$. Besides, the private pools are achieved through indices exclusively owned by blocks, maintaining a unified operation with public pools.

**Initialization.** Based on the shard size and the specified trainable parameter count, we first instantiate the global pools for various linear layer types, respectively. Following standard LoRA practices, we initialize $\mathbf{B}^{pub}$ and $\mathbf{B}^{pri}$ to zero, ensuring consistency with the pretrained model at the start of finetuning. As adopted in PRoLoRA (Wang et al., 2024b), we adjust the sampling boundaries of $\mathbf{A}^{pub}$ and $\mathbf{A}^{pri}$ to align with the vanilla LoRA. Subsequently, we randomly initialize the index matrices $\mathbf{I}_a^k$ and $\mathbf{I}_b^k$, and ensure the private shards are sampled only once. The training and inference procedures remain identical to those of LoRA, except that all low-rank matrices are constructed from the index matrices and global pools.

**Unity.** Even if all the differentiation strategies are elaborated individually for clarity, they can be merged as a unified solution seamlessly for easy implementation. Briefly, MoS utilizes the routing mechanism to retrieve and concatenate shards into low-rank matrices from global pools. Additionally, we justify the theoretical motivation of each differentiation strategy from the perspective of combinational diversity in Appendix. B.1.

## 3.6 Advantage Analysis

Compared to LoRA, MoS only modifies the source of low-rank matrices to concatenated shards from the global pools. This adjustment retains the advantages of LoRA, such as **low-cost switching** and **linear properties**. The former allows for swapping only the finetuned weights instead of all parameters, facilitating the efficient handling of multiple customized models simultaneously. The latter enables MoS to be merged with pretrained weights during inference, thus eliminating inference latency. Moreover, this source modification allows MoS to leverage **existing technical support** of LoRA (Sheng et al., 2023; Chen et al., 2023) with minimal effort. Additionally, MoS offers several extra advantages over LoRA and other peer methods.

**High Parameter Efficiency.** By integrating both inter-layer and intra-layer sharing, MoS demonstrates significantly higher parameter efficiency compared to LoRA and other baselines, as shown

---

[3]https://en.wikipedia.org/wiki/Shard_(database_architecture)

in Sec. 4.2. This reduction in parameters results in decreased disk space and GPU memory usage, thereby greatly alleviating the serving burden in multi-LoRA scenarios.

**High Representation Capacity.** As shown in Table 2, even with a rank of 256 or higher, Tied LoRA and VeRA still clearly underperform other methods. In contrast, the performance of MoS can approximately converge to that of full finetuning as do LoRA and PRoLoRA. This ensures a substantial model capacity, which is crucial for tackling challenging tasks.

# 4 EXPERIMENTS

## 4.1 GENERAL SETUP

Our primary experiments focus on evaluating the instruction following abilities of models, aligning closely with the general configurations outlined by Tulu (Wang et al., 2023) and PRoLoRA (Wang et al., 2024b). Similarly, we adopt a comprehensive assessment protocol that involves factual knowledge, reasoning abilities, multilingual proficiency, and coding skills. Following Wang et al. (2024b), we select specific settings that have demonstrated positive results as evidenced by Table 7 of Wang et al. (2023). Besides, we inherit their unified chatbot framework. This necessitates models to learn both specific tasks and the interaction style. The core configurations are summarized below, with further details available in the Appendix A.

**Datasets.** We finetune base models on Super-Natural Instructions (SuperNI (Wang et al., 2022)) dataset, and evaluate them on both Massive Multitask Language Understanding (MMLU (Hendrycks et al., 2021)) and TyDi QA (Clark et al., 2020) datasets for **factual knowledge** and **multilingual** capabilities, respectively. For the general and mathematical **reasoning** abilities, we finetune models on Flan V2 and its CoT split (Longpre et al., 2023), and conduct evaluation on Big-Bench-Hard (BBH (Suzgun et al., 2022)) and the test set of Grade School Math (GSM8K (Cobbe et al., 2021)) corpora, respectively. Moreover, **coding** skills are evaluated on HumanEval (Chen et al., 2021) dataset after models are finetuned on CodeAlpaca (Chaudhary, 2023) dataset.

**Baselines.** From the perspective of parameter sharing, we evaluate our approach against LoRA and other existing intra-layer or inter-layer sharing methods. A brief introduction to these baselines is provided below.

- **LoRA** (Hu et al., 2021) freezes the pretrained weights and decomposes the weight updates into two trainable low-rank matrices, as formulated in Sec. 2. Following the settings of QLoRA (Dettmers et al., 2023), LoRA is applied to every linear layer across all Transformer blocks, including query, key, value, output, up, gate, and down projection weights.

- **VeRA** (Kopiczko et al., 2023) also freezes pretrained weights, along with randomly initialized shared low-rank matrices, and employs decoupled scaling vectors for the differentiation of weight updates. Consistent with LoRA, VeRA is also applied to all linear layers.

- **Tied LoRA** (Renduchintala et al., 2023) ties the down projection matrices across all query, key, and value projection weights. It shares trainable low-rank matrices across Transformer blocks for parameter savings, while updating the separate scaling vectors for differentiation.

- **PRoLoRA** (Wang et al., 2024b) recently provides an intra-layer solution aimed at enhancing parameter efficiency. It replicates the sub-matrices of LoRA multiple times before differentiating them with partial rotation operations.

## 4.2 MAIN RESULTS

The main results are summarized in Table 2, showcasing the performance of various methods across multiple instruction tuning tasks. Notably, as the rank increases from 2 to 64, the performance of LoRA consistently improves across all benchmarks, significantly surpassing that of the vanilla model. This observation suggests that the current configurations do not introduce apparent parameter redundancy in trainable parameters, highlighting the effectiveness of the setups in demonstrating the parameter efficiency of different approaches.

**In a scenario with a low-level trainable parameter count, MoS outperforms all baseline approaches.** Specifically, to highlight the differences in parameter efficiency among various methods, we fix the number of trainable parameters at a low level (i.e., 5.00M, equivalent to that of LoRA with

Table 2: Results of LLaMA2-7B across multiple instruction-following datasets using different methods. The abbreviations "# Param." and "Avg." refer to "Parameter Count" and "Average", respectively, while the symbols "-sp", "-vs", and "-pd" indicate the ablation of shard privatization, vector sharding, and pair dissociation. The notations "4/8" and "16/32" correspond to increasing the rank to 4 or 8, and 16 or 32, respectively[4]. "$*$" denotes the optional higher ranks. Underlined values indicate the best performance with 5.00M trainable parameters, while bold values denote the best results across all configurations. "†" indicates the results copied from Wang et al. (2024b).

| Method | Rank | # Param. | MMLU (factuality) | BBH (reasoning) | GSM8K (reasoning) | TyDi QA (multilinguality) | | HumanEval (coding) | Avg. |
|---|---|---|---|---|---|---|---|---|---|
| | | | EM (0-shot) | EM (3-shot) | EM (8-shot, CoT) | F1 (1-shot, GP) | EM | P@1 (0-shot) | |
| Vanilla (chat)† | - | - | 41.18 | 0.00 | 3.03 | 17.40 | 0.10 | 0.64 | 10.39 |
| Vanilla (no-chat)† | - | - | 41.53 | 33.43 | 15.47 | 49.18 | 35.35 | 13.57 | 31.42 |
| LoRA | 2† | 5.00M | 44.77 | 36.22 | 26.28 | 48.67 | 35.70 | 18.24 | 34.98 |
| | 8† | 19.99M | 46.55 | 36.92 | 31.11 | 50.50 | 36.89 | 19.37 | 36.89 |
| | 16† | 39.98M | 46.70 | 36.43 | 31.34 | 50.97 | 37.64 | 18.73 | 36.97 |
| | 64 | 159.91M | **47.10** | 37.78 | **31.43** | 51.65 | 38.07 | 19.12 | 37.53 |
| VeRA† | 256 | 1.42M | 42.51 | 35.10 | 22.69 | 48.39 | 36.38 | 18.90 | 34.00 |
| Tied LoRA† | 280 | 4.99M | 44.36 | 35.76 | 25.47 | 50.16 | 37.15 | 18.68 | 35.26 |
| PRoLoRA† | 4/8 | 5.00M | 45.85* | 36.45* | 27.57 | 49.94* | 36.59* | 19.75* | 36.03 |
| MoS | 4/8 | 5.00M | 46.09 | 37.29 | 28.43 | 50.21* | 37.19* | 19.12 | 36.39 |
| | 16/32 | 19.99M | 47.01* | **37.79** | 30.93 | **51.71** | **38.34** | **20.00*** | **37.63** |
| MoS⁻ˢᵖ | 16/32 | 19.99M | 46.64* | 36.69 | 30.17 | 50.27 | 36.90 | 18.60* | 36.54 |
| MoS⁻ᵛˢ | 16/32 | 19.99M | 46.47* | 37.52 | 31.77 | 50.90 | 37.98 | 18.69* | 37.22 |
| MoS⁻ᵖᵈ | 16/32 | 19.99M | 46.23* | 36.17 | 30.71 | 51.40 | 37.94 | 16.77* | 36.54 |

a rank of 2), and compare their performance. Naturally, more superior performance indicates higher parameter efficiency. As shown in Table 2, both the inter-layer shared Tied LoRA and the intra-layer shared PRoLoRA demonstrate better average performance than LoRA. In contrast, by incorporating both intra-layer and inter-layer sharing, our method, MoS, surpasses all of them on average, and achieves consistent improvements across all individual tasks, as indicated by the underlines. However, similar to PRoLoRA (Wang et al., 2024b), we do not claim to achieve greater parameter efficiency than VeRA, given its lower parameter count. However, we attempted to increase the rank of VeRA to match the identical trainable parameter count, but failed due to an out-of-memory (OOM) error. This indicates that VeRA has a very low performance-to-rank ratio, making it impractical due to significantly higher training costs for numerous customization.

**In a regular setting, MoS achieves around $8\times$ savings in trainable parameters.** Considering the significance of performance on various benchmarks, we turn to evaluate the parameter savings of MoS with a performance target, where fewer trainable parameters also indicate high efficiency. We set the performance target to align with LoRA at a rank of 64, which yields an average performance of 37.53. In comparison, with 19.99M trainable parameters (equivalent to that of LoRA with a rank of 8), MoS outperforms LoRA on four out of six metrics, while maintaining comparable average performance. This reconfirms the superior parameter efficiency of MoS, and demonstrates its around eightfold savings in trainable parameters. This can considerably alleviate the burden on service providers, when multiple LoRA-based customizations are served simultaneously.

## 4.3 SCALABILITY ANALYSIS

To assess the robustness of MoS regarding model scale, we extend the experiments to the LLaMA2-13B model. However, we find that finetuning with vanilla LoRA does not yield consistent improvements on the TyDi QA and Code benchmarks. We hypothesize that this phenomenon might be

---

[4]For fair comparison, we keep an identical trainable parameter budget for both LoRA, PRoLoRA and MoS, treat the raised rank of PRoLoRA and MoS as a hyper-parameter, and report the results in one line for more intuitive comparison.

Table 3: Results of LLaMA2-13B across multiple instruction-following datasets using different methods. The abbreviations, notions, and settings align with those of Table 2, respectively.

| Method | Rank | MMLU | BBH | GSM | Avg. |
|---|---|---|---|---|---|
| Vanilla (chat)[†] | - | 50.05 | 0.00 | 2.12 | 17.39 |
| Vanilla (no-chat)[†] | - | 52.04 | 39.67 | 27.82 | 39.84 |
| LoRA[†] | 2 | 52.78 | 42.50 | 36.47 | 43.92 |
| PRoLoRA[†] | 4/8 | 53.27 | 43.12 | 38.72 | 45.04 |
| MoS | 4/8 | **53.51** | **43.65** | **40.79** | **45.98** |

caused by the more powerful capabilities of the base model, which diminishes the effectiveness of the training set. Hence, we exclude these tasks from our analysis.

As listed in Table 3, we impose a limited trainable parameter count (equivalent to that of LoRA with the rank of 2) to compare the representational capacities of different methods. Specifically, compared to the pretrained model, LoRA achieves a notable performance improvement with an average score of 43.92. Building on this, PRoLoRA boosts the average performance by 1%. In contrast, MoS further unleashes the potential of parameters. It consistently exhibits performance enhancements across all individual benchmarks, with another increase of nearly 1% on average. Targeting smaller scale, we also extend the experiments to the LLaMA3.2-3B model with additional random seeds in Appendix. B.3, demonstrating stable performance enhancements across all individual tasks. The considerable improvement reinforces MoS's positive impact on parameter efficiency and its robustness with respect to model sizes.

## 4.4 ABLATION STUDY

In this section, we conduct an ablation study on each differentiation strategy incorporated by MoS, before comparing their relative significance. As discussed in Sec. 2 and listed in Table 1, subset selection plays an indispensable role in reversing the detrimental effects of pure sharing. Since it lays the groundwork for other strategies, we will not reiterate its importance here. Results from the ablation of the other components are presented in Table 2.

**Pair Dissociation.** Rather than using distinct indices for enhanced combination diversity, we apply the same index matrix for the low-rank matrix pair $\mathbf{A}^k$ and $\mathbf{B}^k$ within each linear layer (*i.e.*, $\mathbf{I}_a^k = \mathbf{I}_b^k$). This results in an average performance decline exceeding one percent, emphasizing the critical role of pair dissociation and diverse combinations in improving model performance and aligning with our initial motivation.

**Vector Sharding.** When the basic units in the global pools are reverted back to the row or column vectors of the low-rank matrices from smaller shards, only a 0.41% average decrease in performance is observed. This decline is much less than that associated with pair dissociation. Given that both components aim to enhance combination diversity, these findings imply that pair dissociation provides a more substantial boost to diversity, whereas sharding yields only marginal improvements.

**Shard Privatization.** Different from the previous components, shard privatization enhances differentiation among low-rank matrices by allowing certain shards to be exclusively accessible to one low-rank matrix. Its ablation also leads to a performance drop exceeding one percent, highlighting the significance of privatization.

In summary, all differentiation strategies contribute to improving the parameter efficiency of MoS, despite all being nearly cost-free operations. Among them, pair dissociation and shard privatization unleash the efficiency more saliently through increased combination diversity and exclusive differentiation, respectively, while vector sharding offers only incremental gains in diversity.

## 5 RELATED WORK

**Low-Rank Adaptation.** Low-rank adaptation (LoRA), introduced in Hu et al. (2021), reparameterizes weight updates with two trainable low-rank matrices while the pretrained weights remain fixed. This approach has since spurred numerous enhancements aimed at improving its effectiveness and efficiency (Zhang et al., 2023; Wang et al., 2024a;b). AdaLoRA (Zhang et al., 2023), leveraging

singular value decomposition, automates rank allocation by adaptively pruning vector pairs during finetuning. However, the rank variability across layers complicates the simultaneous serving of multiple LoRAs (Sheng et al., 2023). Closely aligning with our work, VeRA (Kopiczko et al., 2023) implements inter-layer sharing of two frozen random matrices while updating disentangled combination vectors for each layer. However, it necessitates an excessively high rank due to limited model capacity. Subsequently, though Tied LoRA (Renduchintala et al., 2023) alleviates this constraint by allowing shared matrices to be trainable, the tying mechanism on down projection matrices restricts its applicability to linear layers of different dimensions. More recently, PRoLoRA (Wang et al., 2024b) demonstrates enhanced parameter efficiency by reusing sub-matrices within a single linear layer, thus avoiding the above limitations. However, its narrow focus on intra-layer sharing restricts its performance potential. Overall, these approaches typically assume vector pairs as the fundamental units for granted, and concentrate exclusively on inter-layer or intra-layer sharing, both of which limits the potential for parameter efficiency. In contrast, our method utilizes fine-grained shards as basic units and facilitates global sharing to achieve improved parameter efficiency.

**Parameter Sharing.** Parameter sharing has been extensively studied to minimize model sizes. Universal Transformer (Dehghani et al., 2018) shares all layers of a Transformer model, before Takase & Kiyono (2023) further optimizes this approach with three parameter-sharing techniques across Transformer layers for higher efficiency. DictFormer (Lou et al., 2021) implements a more compact model with a shared dictionary, unshared coefficients and indices. For on-device deployment, Edge-Former (Ge et al., 2022) shares attention and FFN modules, and incorporates PEFT-based layer adaptation to reduce parameter count. Pires et al. (2023) eliminates the decoder layer FFN and shares a single, larger FFN across the encoder, achieving notable improvements in accuracy and latency. Recently, Cao et al. (2024) proposes two memory-efficient methods for parameter sharing in attention heads. Differently, our work is centered on the parameter sharing in LoRA modules, and explore the foundational principles guiding the development of parameter sharing strategies.

**Mixture of Experts.** Mixture of Experts (MoE) was initially proposed by Jacobs et al. (1991). Since Zadouri et al. (2023) replaces traditional experts with LoRA modules, various efforts have been made to perform better integration (Wu et al., 2024). Aiming at multi-task learning, Mixture-of-LoRAs (MoA) (Feng et al., 2024) integrates multiple domain-specific LoRAs through a routing strategy, improving both individual task performance and rapid domain adaptation. Similarly, MixLoRA (Li et al., 2024) constructs a sparse LoRA-based MoE model by inserting multiple LoRA experts within the FFN modules and employing independent LoRA adapters in the attention modules. Recently, Mixture of Dyadic Experts (MoDE) (Ning et al., 2024) shares a down-projection matrix for different tasks and employs atomic rank-one adapters with routers for more sophisticated task specialization, enhancing the model's multi-tasking capabilities. In contrast, Zhou et al. (2024) present Mixture-of-Experts with Language Priors Routing (MoE-LPR), featuring a two-stage training process to improve multilingual capability without catastrophic forgetting. To the best of our knowledge, we are the first one to apply MoE for parameter savings within a single LoRA module, rather than focusing on multi-task or multilingual settings.

## 6 Conclusion

Targeting higher parameter efficiency via parameter sharing, we investigate high-level sharing principles, highlighting the essential role of differentiation in reversing the detrimental effects of pure sharing. Building on this finding, we propose a more parameter-efficient LoRA-based method called Mixture of Shards (MoS), incorporating both inter-layer and intra-layer sharing schemes, and integrating four nearly cost-free differentiation strategies: subset selection, pair dissociation, vector sharding, and shard privatization. It maintains all the advantages of LoRA while achieving higher parameter efficiency, and effectively avoids the drawbacks of peer parameter-sharing methods. Our insights into parameter sharing and the MoS method may illuminate future developments in more parameter-efficient finetuning techniques, particularly alleviating the burden for service providers handling multiple customized models simultaneously.

## 7 Acknowledgement

This work was supported in part by Hong Kong Innovation and Technology Support Programme Platform Research Project fund (ITS/269/22FP), RGC grants 17204423, 17205824, C7004-22G (CRF), and C5032-23G (CRF).

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

# A EXPERIMENT DETAILS

Our main experiments concentrate on assessing the instruction-following abilities of models, closely aligning with the frameworks established by Tulu (Wang et al., 2023). In a similar vein, we implement a thorough evaluation protocol that encompasses factual knowledge, reasoning skills, multi-lingual capabilities, and programming expertise. Consistent with Wang et al. (2024b), we select specific configurations of training and test datasets that have yielded positive outcomes, as indicated by the Table 7 of Wang et al. (2023). Additionally, we adopt their integrated chatbot framework, which requires models to acquire both specific tasks and interaction styles. The essential configurations are summarized below.

## A.1 DATASET DETAILS

To evaluate various aspects of model capabilities, we perform comprehensive assessments using multiple datasets. Specifically, we finetune models on the Super-Natural Instructions (SuperNI (Wang et al., 2022)) dataset and then measure their performance on the Multitask Language Understanding (MMLU (Hendrycks et al., 2021)) for factual knowledge and on the TyDi QA (Clark et al., 2020) dataset for multilingual abilities. For general and mathematical reasoning evaluation, we retrain the foundation models on the Flan V2 dataset and its CoT split (Longpre et al., 2023). Their corresponding performances are then presented on the Big-Bench-Hard (BBH (Suzgun et al., 2022)) and the Grade School Math (GSM (Cobbe et al., 2021)) test splits, respectively. Additionally, we utilize the HumanEval (Chen et al., 2021) benchmark to assess the coding abilities of models finetuned on the CodeAlpaca (Chaudhary, 2023) dataset.

To standardize the diverse styles and formats, all instruction tuning datasets are converted to a chatbot-style schema. This includes adding two special tokens, `<|user|>` and `<|assistant|>`, before user inputs and assistant (*i.e.*, language model) outputs, respectively. We also introduce a token `` to indicate the end of each utterance or response. During training, only the sequences following the `<|assistant|>` token and before the next `<|user|>` token are used for loss computation. An illustrative example can be found in the Figure 1 of Wang et al. (2023).

**Finetuning Datasets** The finetuning datasets utilized in our study are detailed as follows:

- **SuperNI** (Wang et al., 2023): This corpus encompasses a diverse array of NLP tasks related to instructions, and is provided under the Apache-2.0 license.
- **Flan V2** (Longpre et al., 2023): This dataset consolidates multiple existing NLP datasets, and enhances them with various data augmentations, as outlined in Chung et al. (2022). The resulting dataset is also available under the Apache-2.0 license.
- **CoT** (Longpre et al., 2023): This dataset includes annotations for chain-of-thought reasoning (Wei et al., 2022). Following Wang et al. (2023), we utilize the CoT mixture derived from the Flan V2 dataset.
- **CodeAlpaca** (Chaudhary, 2023): Designed specifically for code generation, this dataset is created using the Alpaca method (Taori et al., 2023), and is released under the Apache-2.0 license.

**Evaluation Datasets.** The evaluation datasets utilized in our study are detailed as follows:

- **MMLU** (Hendrycks et al., 2021): This benchmark assesses models' factual knowledge through a collection of multiple-choice questions across 57 subjects, including STEM, humanities, and social sciences, with varying difficulty levels from elementary to professional. In our evaluation, we report the exact match (EM) score under a zero-shot setting.
- **GSM** (Cobbe et al., 2021): This corpus evaluates multi-step mathematical reasoning, and contains 8.5K high-quality grade school math problems, among which 1K test items are created by human writers. These problems require 2 to 8 steps of elementary arithmetic to solve. We assess models using 8-shot examples and chain-of-thoughts (CoT), reporting the EM score of the final number in the models' responses.
- **BBH** (Suzgun et al., 2022): This suite consists of 23 challenging tasks from BIG-Bench (Srivastava et al., 2022), aiming at evaluating general multi-step reasoning abilities

of language models. These tasks are specifically selected based on prior evaluations on which earlier models do not outperform average human raters. Our evaluation employs 3 official few-shot examples without chain-of-thought (Direct), and reports the EM score.

- **TyDi QA** (Clark et al., 2020): It is a multilingual question-answering dataset featuring 204K question-answer pairs in 11 topologically diverse languages, collected directly in each language. This benchmark evaluates models' multilingual performance. We adopt the gold passage (GP) setting, where the correct answer is provided in a reference passage, utilize one-shot prompting, and report both EM and F1 scores.

- **HumanEval** (Chen et al., 2021): This dataset includes 164 programming problems, each with a function signature, docstring, body, and unit tests, serving as a benchmark for assessing models' coding capabilities by measuring the functional correctness of synthesized programs from docstrings. We report the pass@1 metric using zero-shot prompting with a sampling temperature of 0.1.

## A.2 Hyperparameter Configurations

We utilize LLaMA2-7B, 13B (Touvron et al., 2023), and LLaMA3.2-3B (Dubey et al., 2024) as the foundation models for our experiments, which are conducted on a single NVIDIA A100-40G GPU. Each experimental configuration is repeated two times with the random seed as 0 and 1, respectively, before reporting the average performance. The detailed procedures for finetuning and evaluation are outlined below.

**Finetuning Setup.** To reduce memory cost during finetuning, we follow the approach outlined in QLoRA (Dettmers et al., 2023), loading all pretrained models in a 4-bit NormalFloat format and utilizing a Paged AdamW Optimizer. Specifically, we utilize 4-bit quantized versions of LLaMA2-7B, 13B, and Llama3.2-3B models in our experiments. Next, we apply LoRA to all linear layers within the Transformer blocks, including the query, key, value, output, up, gate, and down projection weights, setting the scaling factor $\alpha$ to 16 and the dropout rate to 0.1. For more efficient finetuning, we also follow the configuration from Wang et al. (2024b) to set a batch size of 16 and the maximum sequence length to 512, truncating samples during preprocessing if needed. We also cap the maximum gradient norm at 0.3 to enhance training stability. LLaMA2-7B and 13B undergo 10,000 steps of finetuning using a linear learning rate scheduler with a warmup ratio of 3%, while LLaMA3.2-3B is finetuned for one epoch in each task. Additionally, we search for the optimal learning rate for LoRA, and apply this value to both LoRA and MoS. Specifically, with a LoRA rank of 8, we search for the best learning rate from {2e-5, 5e-5, 1e-4, 2e-4, 5e-4, 1e-3, 2e-3}. Our preliminary experiments demonstrate that 2e-4 performs the best.

**Evaluation Setup.** For inference, we leverage vLLM (Kwon et al., 2023), an efficient inference and serving library for LLMs, which significantly speeds up the generation process with minimal impact on performance. We also employ greedy decoding with a maximum length of 512. For the details of scoring calculations, we closely follow the GitHub repository of TULU[5].

## B Additional Analysis

### B.1 Theoretical Motivations of Differentiation Strategies

As shown in Sec. 2, differentiation reverses the detrimental effects of the the pure sharing mechanism. Here we demonstrate the design motivation of each specific differentiation strategies under the guidance of combinational diversity. Specifically, we approximately conceptualize differentiation as the combinational diversity (*i.e.*, number of potential combinations) of each low-rank matrix pair. Aligning with the notations in Sec. 3, let $L$ denote the number of Transformer blocks, $r$ the rank, $e$ the equivalent rank of LoRA in terms of trainable parameters, and $l$ the number of shards per vector. In the case of pure sharing, the potential combinations for each low-rank matrix pair can be expressed as $C_{Le}^{Le} = 1$, since all the parameters are shared in the same way for each pair. Subset selection enhances the number of combinations to $C_r^{Le}$ by selecting a subset of vector pairs for each

---

[5]https://github.com/allenai/open-instruct

low-rank matrix pair. Pair dissociation rapidly increases the combinational diversity by separating vector pairs into distinct pools, allowing for independent selection of vectors for each matrix in a low-rank pair, resulting in a combination count of $C_r^{Le} \times C_r^{Le}$. Vector sharding, which breaks down vectors into smaller shards, further amplifies the number of combinations to $C_{rl}^{Lle} \times C_{rl}^{Lle}$, given that $C_r^{Le} < C_{rl}^{Lle}$ when $r < Le$ and $l > 1$. Compared to LoRA, pure sharing shares all the parameters across layers, resulting in performance degradation. Shard privatization, however, partially reverses this trend by dividing the global pool into public and private sections for improved differentiation.

## B.2 FURTHER EXPERIMENTS ON DIFFERENTIATION

Compared to the pure sharing baseline, random scaling only introduces noise to the initialization of scalers, resulting in slight performance improvement. As a more aggressive measure, subset selection randomly masks specific vector pairs, and outperforms pure sharing remarkably. To further confirm this conclusion, we also extend these experiments to the LLaMA3.2-3B model (Dubey et al., 2024). As detailed in Table 4, random scaling still exhibits slight improvement over pure sharing, while subset selection boosts the performance with a margin over 0.8% on average.

Table 4: Results of LLaMA3.2-3B with different sharing and differentiation methods across diverse instruction following datasets. "+ Random Scaling" and "+ Subset Selection" denote the individual integration of them into the "Pure Sharing" scheme, respectively.

| Method | Rank | # Param. | MMLU | BBH | GSM8K | TyDi QA | | HumanEval | Avg. |
|---|---|---|---|---|---|---|---|---|---|
| | | | EM | EM | EM | F1 | EM | P@1 | |
| LoRA | 2 | 3.04M | 51.97 | 37.07 | 33.74 | 62.59 | 44.39 | 31.17 | 43.49 |
| Pure Sharing | 56 | 3.04M | 51.25 | 38.73 | 31.92 | 61.86 | 45.17 | 30.46 | 43.23 |
| + Random Scaling | 56 | 3.04M | 51.35 | 38.81 | 32.75 | 62.18 | 45.11 | 30.52 | 43.45 |
| + Subset Selection | 56 | 3.04M | 51.86 | 39.88 | 33.89 | 62.28 | 45.31 | 31.14 | 44.06 |

## B.3 ROBUSTNESS ANALYSIS

**Performance Robustness.** We further repeat the experiments with 4 random seeds (*i.e.*, 0, 1, 2, 3) and the LLaMA3.2-3B model. As shown in Table 5, MOS with an equivalent parameter budget to LoRA with the rank of 8 performs competitively with LoRA with the rank of 64. This validates an $8\times$ parameter reduction achieved by MOS again, aligning with our previous conclusions. Meanwhile, compared to LoRA, MOS exhibits comparable (even lower on average) standard deviations, and provides a higher average value, indicating its similar stability and better performance.

**Hyperparameter Robustness.** To demonstrate the robustness of hyperparameters, we perform a grid search for the configurations of private rank and shard number per vector with 4 seeds (*i.e.*, 0, 1, 2, 3) on the LLaMA3.2-3B model and BBH benchmark. As listed in Table 6, as the number of shards increases to provide more differentiation, the optimal private rank tends to decrease. Generally, shard numbers of 4 or 8 yield the best results. From another perspective, for any given private rank, there always exists a suitable range of shard numbers that consistently produce remarkable results (*i.e.*, $\geq$ 39.8%), demonstrating the robustness of private rank.

Table 5: Results of LLaMA3.2-3B across multiple instruction-following datasets using different methods. The abbreviations "# Param." and "Avg." refer to "Parameter Count" and "Average", respectively, while the subscripts denote the standard deviations.

| Method | Rank | # Param. | MMLU | BBH | GSM8K | TyDi QA | | HumanEval | |
|---|---|---|---|---|---|---|---|---|---|
| | | | EM | EM | EM | F1 | EM | P@1 | Avg. |
| LoRA | 8 | 12.16M | $52.35_{\pm0.25}$ | $38.74_{\pm0.89}$ | $37.19_{\pm1.59}$ | $63.46_{\pm1.06}$ | $46.09_{\pm1.14}$ | $30.94_{\pm0.21}$ | $44.79_{\pm0.86}$ |
| | 64 | 97.26M | $52.36_{\pm0.23}$ | $39.70_{\pm0.52}$ | $37.98_{\pm0.72}$ | $64.04_{\pm1.51}$ | $46.53_{\pm1.89}$ | $31.83_{\pm0.21}$ | $45.41_{\pm0.85}$ |
| MOS | 16 | 12.16M | $52.41_{\pm0.17}$ | $40.14_{\pm1.01}$ | $37.89_{\pm0.71}$ | $63.78_{\pm1.05}$ | $46.31_{\pm1.21}$ | $31.78_{\pm0.23}$ | $45.38_{\pm0.73}$ |

**Significance Test.** Furthermore, we conduct the significance test between LoRA and MoS on Rank = 2 and 8 across all benchmarks in our main experiments on the LLaMA2-7B model. As shown in Table 7, the p-values between LoRA and MoS with both high and low trainable parameter budgets are smaller than 5%, indicating the statistical significance of our results.

Table 6: Results of MoS with the LLaMA3.2-3B model and BBH benchmark across different shards per vector and private ranks.

| Shards per Vector | Private Rank | | | |
|:---:|:---:|:---:|:---:|:---:|
| | 1 | 3 | 5 | 7 |
| 1 | 38.9 | 39.3 | 39.1 | 39.3 |
| 2 | 38.6 | 39.3 | 39.7 | 39.5 |
| 4 | 39.3 | 39.8 | 40.0 | 39.6 |
| 8 | 39.6 | 39.8 | 39.6 | 38.8 |
| 16 | 39.8 | 39.3 | 39.3 | 38.6 |

Table 7: P-values of the significance test of LoRA and MoS on the LLaMA2-7B model across two different trainable parameter budgets.

| Method | Rank | # Param. | p-value |
|:---|:---:|:---:|:---:|
| LoRA vs. MoS | 2 | 5.00M | 0.03% |
| LoRA vs. MoS | 8 | 19.99M | 1.29% |

## C  LIMITATIONS

**Slightly more GPU consumption is required for finetuning.** Compared to LoRA with the same trainable parameter count, MoS increases the rank several times, leading to similar GPU consumption to LoRA with the same rank. However, MoS does provide better performance. We also finetune the LLaMA3.2-3B model for one epoch across various tasks, and record the time consumption of LoRA and MoS in Table 8. It is worth noticing that MoS only incurs 2.80% more finetuning time.

**The routing mechanism may incur more inference latency.** Targeting this drawback, we give up existing activation-based routing mechanisms, because the routing operation has to wait for the activations, resulting in larger latency. Instead, we intentionally adopt the index-based routing mechanism so that precomputing can be used to prepare the whole low-rank matrices in parallel to the computation of preceding transformer blocks. This could circumvent the latency of routing operation, and apply all existing inference techniques of LoRA.

Table 8: Comparison of fine-tuning time consumption (in hours) between LoRA and MoS[6].

| Method | Rank | # Param. | MMLU | BBH | GSM8K | Codex-Eval | Avg. |
|:---|:---:|:---:|:---:|:---:|:---:|:---:|:---:|
| LoRA | 8 | 12.16M | 1.50 | 1.47 | 1.82 | 0.21 | 1.25 |
| MoS | 8 | 12.16M | 1.54 | 1.52 | 1.86 | 0.22 | 1.29 |

---

[6]Due to the same finetuning process as MMLU benchmark, TyDi QA is neglected here to avoid repetitive computation.

