# OpenReview forum: "MoS: Unleashing Parameter Efficiency of Low-Rank Adaptation with Mixture of Shards"
_ICLR.cc/2025/Conference — ICLR 2025 Poster_

### Official Review · Reviewer_doAZ · 2024-11-03

**Soundness:** 3
**Presentation:** 3
**Contribution:** 3
**Rating:** 6
**Confidence:** 3

**Summary:**

This paper proposes Mixture of Shards (MOS), a LoRA-based method designed to reduce trainable parameters while maintaining performance for LLMs. MOS combines inter and intra layer sharing mechanisms with MoE-like routing system for parameter selection. It also introduces four “differentiation strategies”: subset selection, pair dissociation, vector sharding, and shard privatization (to add diversity and prevent performance degradation from parameter sharing). The authors claim that MOS achieves about an 8x reduction in parameters compared to LoRA while retaining competitive performance.

MOS method proposes leveraging both VeRA-like inter-layer parameter sharing and PRoLoRA-like intra-layer parameter sharing. It is proposing “Global Sharing Scheme” where each adapted layer across the Transformer creates its low-rank matrices (A and B) using shards from a globally shared pool selected by MoE-like routing.

“Differentiation Strategies” used in MOS:

-Subset selection - selects a subset of vector pairs per transformer block

-Pair dissociation - separates vector pairs into two different pools to create unique combinations for each block.

-Vector sharding - breaks down vectors into smaller shards, which are then concatenated.

-Shard privatization - divides the global pool into public and private sections.

**Strengths:**

-The paper has solid motivation.

-Authors propose an intuitively sound idea of using a combination of inter layer and intra layer sharing with MoE-like routing.

-The authors claim that MOS is the first method to use an MoE-like mechanism for parameter-efficient fine-tuning in a single-task LoRA.

-The comparisons with LoRA, VeRA, and PRoLoRA are relevant baselines for MOS performance.

-I like the design of the initial experiment (Table 1) - it's good to back up and motivate the method (though I have some comments mentioned in the weaknesses).

**Weaknesses:**

I think that the idea behind MOS is interesting and worth exploring, but it needs more rigorous experiments and analysis, along with a clearer explanation of the method and design choices. Without the code, reproducibility is challenging.

I would like to understand why we would select this method over simpler methods like VeRA that provide good results. Does this method offer enough benefits to justify its complexity? The complexity should be justified, and any impact on finetuning overhead should be mentioned.

Weaknesses:

-The motivation for the specific “differentiation” strategies could be clearer. The authors mention that these strategies help maintain representational power, but this is very high-level and lacks theoretical support.

-The MoE-like routing mechanism for parameter selection isn’t clearly explained, making it hard to reproduce. What exactly is the routing algorithm? Were other approaches tested?

-The paper only evaluates MOS on instruction-following tasks.

-The comparison with VeRA isn’t entirely fair, as MOS uses more parameters than VeRA. I understand that VeRA can have practical limitations in increasing parameters, but could we reduce the MOS parameter count to match VeRA?

-Standard deviations are not provided.

-The initial experiment (Table 1) is interesting, but the conclusion about random scaling doesn’t seem fully justified - this strategy shows very minimal improvement and might not be statistically significant. For subset selection, could more models and seeds be tested to confirm the results?

-Could we add some additional models? Even smaller ones could help validate MOS’s performance. The current results are limited to LLaMA2-7B and LLaMA2-13B, with minimal gains for the latter, which may not justify MOS complexity.

**Questions:**

Questions:

-What’s the reason behind choosing the specific differentiation strategies? How was each expected to impact performance, and was the decision based on empirical results?

-How exactly are vectors selected from the global sharing pool?

-Does the MOS method affect finetuning time? (given the added complexity of MOS)

-Do you plan to release the code? This is a complex framework, and code would be very useful

-Can you elaborate on the statement that differentiation “reverses the detrimental effects of sharing”? Is there any theoretical support for MOS’s design?

-Why was standard deviation not provided for the averages?

-“Differentiation” is typically associated with gradient computation in deep learning, which might cause confusion. I’d consider a different name.

---

> ### Author Response · Authors · 2024-11-25
> **Response to Reviewer doAZ (1/n)**
>
> Dear Reviewer doAZ,
>
> Thank you for the thorough and insightful reviews on our work! We are grateful for the time you spent on our submission. Below, we provide detailed responses to your comments one by one, and hope that they can fully address your concerns!
>
> **Weakness 1: Concern about reproducibility.**
>
> - We have attached the code in the supplementary materials of our submission. If needed, you can check it for details. Besides, we will soon open-source our code and environment for better reproducibility with minimal efforts.
>
> **Weakness 2.1: Comparison with VeRA.**
>
> Targeting at this confusion, we wanna start with the problems we met while implementing VeRA:
>
> - **Low reproducibility.** Until now, VeRA has not been officially open-sourced to reproduce the reported performance. We have to read the paper carefully, and compare with the implementation of Nvidia to ensure capturing all the details [1].
> - **Conflict of representation capacity and training/inference costs.** Despite the claimed higher parameter efficiency, we find that VeRA needs an extremely higher rank for this purpose. As shown in Table 2 of our paper, the average performance of VeRA with the rank of 256 is 34.00, which is even lower than that of LoRA with the rank of 2 (i.e., 34.98). This is far from the best performance. **An extremely high rank is inevitably required for VeRA.** However, **this will incur high training and inference costs.** Otherwise, users have to tolerate inferior performance. We also tried to increase the rank of VeRA for fair comparison. However, even with the equivalent trainable parameter count as LoRA with the rank of 1, VeRA will cause out-of-memory in our NVIDIA A100-40G GPU, making the comparison infeasible and demonstrating the practical inefeasibility of VeRA.
>
> Both of the above drawbacks cause the low feasibility for the practical usage of VeRA. Actually, we **also discussed with other researchers**, and they encountered exactly the same issues, and had to give up the application of VeRA.
>
> **Weakness 2.2: Potential overheads of MoS.**
>
> Honestly, we have already **tried to avoid any apparent drawbacks when designing MoS**, and find the following left overheads:
>
> - **Slightly more GPU consumption for finetuning.** Compared to LoRA with the same trainable parameter count, MoS increases the rank several times, leading to similar GPU consumption to LoRA with the same rank. However, **MoS does provide better performance, and avoids the extremely higher rank of VeRA**, as discussed above. We also finetune the LLaMA3.2-3B model for one epoch across various tasks, and record the time consumption (in hours) of LoRA and MoS in the following table. It is worth noticing that MoS only incur 2.80% more finetuning time. We tried to solve it, but haven't found a good solution for a free launch. If you're interested, we welcome any constructive discussions!
> - **The routing mechanism may incur more inference latency.** Targeting this drawback, we give up existing activation-based routing mechanisms, because the routing operation has to wait for the activations, resulting in larger latency. Instead, we intentionally adopt the index-based routing mechanism so that **precomputing can be used** to prepare the whole low-rank matrices in parallel to the computation of preceding transformer blocks. This could **circumvent the latency of routing operation**, and apply all the existing inference techniques of LoRA.
>
>
> | Method | Rank | Parameters | MMLU | BBH | GSM | Codex-Eval | Avg. |
> | ------ | ---- | ---------- | ---- | --- | --- | ---------- | ---  |
> | LoRA   | 8    | 12.16M     | 1.50 | 1.47 | 1.82 | 0.21       | 1.25   |
> | MoS    | 8    | 12.16M     | 1.54 | 1.52 | 1.86 | 0.22       | 1.285   |

---

> > ### Author Response · Authors · 2024-11-25
> > **Response to Reviewer doAZ (2/n)**
> >
> > **Weakness 3: Motivation for the specific "differentiation" strategies.**
> >
> > - As motivated by the analysis in Sec. 2, we mainly **design the specific differentiation strategies intuitively under the guidance of combinational diversity.** Specifically, we approximately conceptualize differentiation as the combinational diversity (i.e., number of potential combinations) of each low-rank matrix pair. Aligning with the notations in our submission, let $L$ denote the number of Transformer blocks, $r$ the rank, $e$ the equivalent rank of LoRA in terms of trainable parameters, and $l$ the number of shards per vector. In the case of pure sharing, the potential combinations for each low-rank matrix pair can be expressed as $C^{Le}\_{Le} = 1$, since all the parameters are shared in the same way for each pair. Subset selection enhances the number of combinations to $C^{Le}\_r$ by selecting a subset of vector pairs for each low-rank matrix pair. Pair dissociation rapidly increases the combinational diversity by separating vector pairs into distinct pools, allowing for independent selection of vectors for each matrix in a low-rank pair, resulting in a combination count of $C^{Le}\_r \cdot C^{Le}\_r$. Vector sharding, which breaks down vectors into smaller shards, further amplifies the number of combinations to $C^{Lle}\_{rl} \cdot C^{Lle}\_{rl}$, given that $C^{Le}\_r < C^{Lle}\_{rl}$ when $r < Le$ and $l > 1$. Compared to vanilla LoRA, pure sharing shares all the parameters across layers. Shard privatization, however, partially reverses this trend by dividing the global pool into public and private sections for improved differentiation.
> >
> > **Weakness 4: Explanation of routing mechanism.**
> >
> > - As mentioned above, instead of activation-based routing operation, we **select index-based routing mechanism to circumvent the potential inference latency.** Specifically, it equips each low-rank matrix with a small index matrix. During training or inference, all the corresponding shards will be retrieved from the global pools based on the index matrix, before these shards are concatanated into the low-rank matrix.
> > - Due to the extensive variations in routing mechanisms, we do not rule out the possibility of better alternatives. However, **our contribution primarily lies in insight into the significance of differentiation for parameter sharing, which guides us to design MoS with higher parameter efficiency.** While the specifics of the routing mechanism are important, they are not the central focus of our research. For the inference latency concern, we select this index-based routing mechanism directly.
> >
> > **Weakness 5: Only instruction-following tasks.**
> >
> > - Due to the unaffordable costs of pretraining, instruction-tuning is the most common form of LLM finetuning. Despite this, we **intentionally choose diverse tasks, covering factual knowledge, reasoning, multilinguality, and coding**, with the hope of assessing the abilities of MoS as comprehensively as possible.
> > - Besides, methods of LoRA series are quite general, and **mainly impact the representation capacity for finetuning, thereby being robust for task categories.** We welcome any open discussion on the choices of tasks. If you expect other tasks, please feel free to let us know, and we can supplement relevant experiments.

---

> > > ### Author Response · Authors · 2024-11-25
> > > **Response to Reviewer doAZ (3/n)**
> > >
> > > **Weakness 6:  More Models & Providing Standard Deviations.**
> > >
> > > - Due to limited resources, we initially conducted our experiments with two seeds. Based on the results of Table 2, we conduct the significance test between LoRA and MoS. As shown below, the p-values between LoRA and MoS with both high and low trainable parameter budgets are smaller than 5%, **indicating the statistical significance of our results.**
> > > - **For more robust demonstration of standard deviations and the extension on additional models,** we further repeat the experiments with more seeds (i.e., 0, 1, 2, 3) and the LLaMA3.2-3B model, which is one of the latest models and differs in size from those in our submission. As shown in the following table, even without much hyperparameter optimization, the performance of MoS with an equivalent parameter budget to LoRA with the rank of 8 remains competitive with that of LoRA with the rank of 64. This validates an 8x parameter reduction achieved by MoS again, aligning with our previous conclusions. Meanwhile, compared to LoRA, MoS exhibits comparable (even lower on average) standard deviations, and provides a higher average value, indicating **its similar stability and better performance.** These results will be further supplemented into our final paper!
> > >
> > > Significance Test:
> > >
> > >
> > > |    Method    | rank | # Param. | p-value |
> > > |:---:|:----:|:--:|:---:|
> > > | LoRA vs. MoS |   2  |    5.00M   |  0.03%  |
> > > | LoRA vs. MoS |   8  |   19.99M   |  1.29%  |
> > >
> > >
> > > Avg. Performance:
> > >
> > >
> > > | **Method** | **Rank** | **# Param.** | **MMLU**  | **BBH**  | **GSM8K**  | **TyDi QA**  | **TyDi QA** | **HumanEval** | **Avg.** |
> > > |---|--|-|-|---|-|---|---|--|--|
> > > | **LoRA**   | 8        | 12.16M       | 52.35                | 38.74              | 37.19                | 63.46                            | 46.09                            | 30.94                      | 44.79    |
> > > |            | 64       | 97.26M       | 52.36                | 39.70              | 37.98                | 64.04                            | 46.53                            | 31.83                      | 45.41    |
> > > | **MoS**    | 16       | 12.16M       | 52.41                | 40.14              | 37.89                | 63.78                            | 46.31                            | 31.78                      | 45.38    |
> > >
> > >
> > > std:
> > >
> > >
> > > | **Method** | **Rank** | **# Param.** | **MMLU**  | **BBH**  | **GSM8K**  | **TyDi QA**  | **TyDi QA** | **HumanEval** | **Avg.** |
> > > |---|--|-|-|---|-|---|---|--|--|
> > > | **LoRA**   | 8        | 12.16M       | 0.25                  | 0.89                | 1.59                  | 1.06                             | 1.14                             | 0.21                       | 0.86     |
> > > |            | 64       | 97.26M       | 0.23                  | 0.52                | 0.72                  | 1.51                             | 1.89                             | 0.21                       | 0.85     |
> > > | **MoS**    | 16       | 12.16M       | 0.17                  | 1.01                | 0.71                  | 1.05                             | 1.21                             | 0.23                       | 0.73     |
> > >
> > >
> > > **Weakness 7: Confirmation of the conclusion about subset selection.**
> > >
> > > - Compared to the baseline, random scaling only introduces noise to the initialization of scalers, and results in slight improvement. As a more aggressive measure, subset selection randomly masks specific vector pairs, and outperforms pure sharing remarkably. **This trend suggests a pressing need for differentiation in pure sharing**, motivating our choice of subset selection over random scaling in the following design.
> > > - To further confirm this conclusion, we also **extend these experiments to the LLaMA3.2-3B model.** As detailed in the following table, random scaling still exhibits slight improvement over pure sharing, while subset selection boosts the performance by over 0.8% on average.
> > >
> > > Hope that these results can help you confirm the remarkable benefits of subset selection!
> > >
> > >
> > > | **Method** | **Rank** | **# Param.** | **MMLU**  | **BBH**  | **GSM8K**  | **TyDi QA**  | **TyDi QA** | **HumanEval** | **Avg.** |
> > > |---|--|-|-|---|-|---|---|--|--|
> > > | **Pure Sharing**  | 56       | 3.04M        | 51.25                | 38.73               | 31.92                 | 61.86                            | 45.17                            | 30.46                      | 43.23    |
> > > | **+ Random Scaling**         | 56       | 3.04M        | 51.35                | 38.81               | 32.75                 | 62.18                            | 45.11                            | 30.52                      | 43.45    |
> > > | **+ Subset Selection**       | 56       | 3.04M        | 51.86                | 39.88               | 33.89                 | 62.28                            | 45.31                            | 31.14                      | 44.06    |

---

> > > > ### Author Response · Authors · 2024-11-25
> > > > **Response to Reviewer doAZ (4/n)**
> > > >
> > > > **Questions:**
> > > >
> > > > **Q1: Rationales behind differentiation strategies.**
> > > >
> > > > - Please refer to our clarification on "Weakness 3: Motivation for the specific 'differentiation' strategies".
> > > >
> > > > **Q2: Selection of vectors from the global sharing pool.**
> > > >
> > > > - Please refer to our elaboration on routing mechanism in "Weakness 4: Explanation of routing mechanism".
> > > >
> > > > **Q3: Impact on finetuning time.**
> > > >
> > > > - We finetune the LLaMA3.2-3B model for one epoch across various tasks, and record the time consumption (in hours) of LoRA and MoS in the following table. It is worth noticing that MoS does incur 2.80% more finetuning time. As mentioned in "Weakness 2.2: Potential overheads of MoS", this is mainly caused by the increased rank for better performance. We do not find a free lunch here, and welcome any suggestive discussions!
> > > >
> > > >
> > > > | Method | Rank | Parameters | MMLU | BBH | GSM | Codex-Eval | Avg. |
> > > > | ------ | ---- | ---------- | ---- | --- | --- | ---------- | ---  |
> > > > | LoRA   | 8    | 12.16M     | 1.50 | 1.47 | 1.82 | 0.21       | 1.25   |
> > > > | MoS    | 8    | 12.16M     | 1.54 | 1.52 | 1.86 | 0.22       | 1.285   |
> > > >
> > > >
> > > > **Q4: Release the code.**
> > > >
> > > > - Yes. As mentioned in "Weakness 1: Concern about reproducibility.", we have attached the code in the supplementary materials of our submission. If needed, you can check it for details. Besides, we will soon **open-source our code and environment for better reproducibility with minimal efforts.**
> > > >
> > > > **Q5.1: Elaboration on the statement that differentiation "reverses the detrimental effects of sharing".**
> > > >
> > > > - Based on the previous conclusions that VeRA and PRoLoRA achieve better performance than LoRA via parameter sharing with inter-layer and intra-layer sharing respectively, a straightforward way for better performance is to fully share all parameters (i.e., pure sharing). However, we find that **this could result in inferior performance than vanilla LoRA.** Then we try to add simple differentiation measures (i.e., random scaling and subset selection), and find that **the introduction of subset selection can help pure sharing outperform vanilla LoRA**, which we claim that differentiation reverses the detrimental effects of sharing.
> > > >
> > > > **Q5.2: Theoretical support for MoS's design.**
> > > >
> > > > - For the theoretical support for MoS’s design, please refer to our clarification in "Weakness 3: Motivation for the specific 'differentiation' strategies".
> > > >
> > > > **Q6: Providing Standard Deviations.**
> > > >
> > > > - Please refer to our clarification in "Weakness 6:  More Models & Providing Standard Deviations".
> > > >
> > > > **Q7: Confusing name of "Differentiation".**
> > > >
> > > > - Thank you so much for pointing out this confusing wording! We will consider it seriously for a more suitable name!
> > > >
> > > > BTW, although four differentiation strategies are integrated into MoS for higher parameter efficiency, which may seem to complicate the method, all of these strategies are nearly cost-free, and can be merged seamlessly for easy implementation. **Briefly, MoS utilizes an index matrix to retrieve and concatenate shards into each low-rank matrix.** Hope that this can further justify its complexity!
> > > >
> > > > Thanks for your time devoted to our work again! We hope that these explanations could help you further understand the importance of differentiation and its guidance for our method, and you can reconsider our scores accordingly! If you have any further questions, please do not hesitate to reach out; we welcome any insightful discussions!
> > > >
> > > > Sincerely,
> > > >
> > > > Authors
> > > >
> > > >
> > > > **References**
> > > >
> > > > [1] https://github.com/NVIDIA/NeMo/tree/adithyare/vera

---

> > > > > ### Comment · Reviewer_doAZ · 2024-11-26
> > > > >
> > > > > Thank you for responding and addressing my questions and concerns. I have decided to increase the score based on the understanding that clarification and new results will be added to the work.

---

> ### Author Response · Authors · 2024-11-26
> **Response to Reviewer doAZ**
>
> Dear Reviewer doAZ,
>
> Thank you again for your thoughtful comments, acknowledgment of our work, and reconsideration of our scores! In particular, your feedback has significantly contributed to further polishing our work. **We promise you that all these results and valuable discussions will be supplemented into our final paper, and our code will also be open-sourced for better reproducibility!** Thanks!
>
> Sincerely,
>
> Authors

---

### Official Review · Reviewer_Gvvt · 2024-11-04

**Soundness:** 3
**Presentation:** 3
**Contribution:** 2
**Rating:** 6
**Confidence:** 4

**Summary:**

This paper introduces Mixture of Shards (MoS), a sharded adaptation of LoRA designed to achieve greater parameter efficiency by leveraging parameter sharing across layers and within layers. MoS not only reduces the number of parameters required compared to traditional LoRA but also mitigates the potential performance degradation associated with excessive sharing. This is achieved through four strategies: subset selection, pair dissociation, vector sharding, and shard privatization, which ensure that each shared parameter can adapt to specific model requirements. MoS demonstrates a further reduction in trainable parametric usage, allowing more scalable deployment of LoRA-based models.

**Strengths:**

1. MoS combines subset selection, pair dissociation, vector sharding, and shard privatization to reduce parameters while maintaining performance.
2. Demonstrates an eightfold parameter reduction compared to traditional LoRA with empirical support.
3. Provides insights into the contributions of each differentiation strategy.

**Weaknesses:**

1. Although the paper introduces subset selection, it lacks criteria for choosing subsets; the selection is randomly initialized and fixed throughout training.
2. The MoS approach is primarily a combination of various techniques rather than a cohesive, unified method.
3. The MoS approach introduces significant randomness, making it challenging to determine if the improvements result from the design or from random variations. A test of significance could strengthen these claims.
4. The paper includes limited ablation studies for MoS, making it difficult to isolate and understand the contributions of each individual strategy in the overall design.

**Questions:**

1. For the experiments conducted with two runs using seeds 0 and 1, could you provide the individual performance results for each run? Additionally, were any further experiments conducted with different seeds to assess the robustness of the results?
2. How does the random initialization impact the performance of MoS? Given the reliance on randomness, are there specific initialization settings or hyperparameters that consistently yield better results?
3. What criteria, if any, were used to decide the number of shards in the global pool, and how sensitive is the model’s performance to this choice?
4. Were there any specific cases where certain differentiation strategies (e.g., subset selection, pair dissociation) proved more beneficial than others?
5. How does the computational overhead of MoS compare to traditional LoRA during training and inference, especially with regard to memory usage and GPU hours?
6. Since MoS integrates multiple strategies, are there any known trade-offs between parameter savings and performance across tasks?

---

> ### Author Response · Authors · 2024-11-25
> **Response to Reviewer Gvvt (1/n)**
>
> Dear Reviewer Gvvt,
>
> Thanks for your dedication to our reviewing process and your recognition on our work! We will try our best to clarify your comments one by one below. Hope that we can address your concerns!
>
> **Weakness 1: Lack of design for subset selection.**
>
> - Actually, **an inappropriate routing mechanism may incur more inference latency.** Targeting this drawback, we give up existing activation-based routing mechanisms, because the routing operation has to wait for the activations, resulting in larger latency. Instead, we intentionally adopt the index-based routing mechanism so that **precomputing can be used** to prepare the whole low-rank matrices in parallel to the computation of preceding transformer blocks. This could **circumvent the latency of routing operation, and apply all the existing inference techniques of LoRA.**
> - Due to the extensive variations in routing mechanisms, we do not rule out the possibility of better alternatives. However, **our contribution primarily lies in insight into the significance of differentiation for parameter sharing, which guides us to design MoS with higher parameter efficiency.** While the specifics of the routing mechanism are important, they are not the central focus of our research. For the inference latency concern, we select this index-based routing mechanism directly.
>
> **Weakness 2: Lack of cohesion and unity.**
>
> In our paper, we elaborate four differentiation strategies individually **for clarity**, and highlight their relationship with our first contribution (i.e., the analysis of differentiation for parameter sharing).
>
> Actually, **all of these strategies are nearly cost-free, and can be merged seamlessly for easy implementation.** Briefly, MoS utilizes the routing mechanism to retrieve and concatenate shards into each low-rank matrix from global pools. Hence, we still believe that MoS is cohesive and unified. Hope that this can further justify its unity!
>
> **Weakness 3: Lack of the test of significance.**
>
> We fully understand your concern about the robustness of MoS. For your reference, we conduct the significance test between LoRA and MoS. As shown below, the p-values between LoRA and MoS with both high and low trainable parameter budgets are smaller than 5%, indicating the statistical significance of our results.
>
> |    Method    | rank | # Param. | p-value |
> |:------------:|:----:|:----------:|:-------:|
> | LoRA vs. MoS |   2  |    5.00M   |  0.03%  |
> | LoRA vs. MoS |   8  |   19.99M   |  1.29%  |
>
>
> **Weakness 4: Insufficient ablation study.**
>
> As presented in Sec.4.4 of our submission, we conduct an ablation study on differentiation strategies,  including shard privatization, vector sharding, and pair dissociation, and compare their relative significance. For subset selection, as discussed in Sec. 2 and listed in Table 1, it plays an indispensable role in reversing the detrimental effects of pure sharing. Since it lays the groundwork for other strategies, we do not reiterate its importance in Sec. 4.4. **With all the analysis, we guess all the components have been ablated, and not that sure about extra ablation study. If you have any suggestions, please do not hesitate to let us know!**

---

> > ### Author Response · Authors · 2024-11-25
> > **Response to Reviewer Gvvt (2/n)**
> >
> > **Question 1: Robustness of results.**
> >
> > We fully understand your concerns about the robustness of our results.
> >
> > - Firstly, as discussed in "Weakness 3: Lack of the test of significance", the p-values between LoRA and MoS **indicate the statistical significance of our current results.**
> > - For more robust demonstration, we **further repeat the experiments with more seeds (i.e., 0, 1, 2, 3) and the LLaMA3.2-3B model,** which requires less computing resources, and is requested by other reviewers for an additional model. As shown in the following table, even without much hyperparameter optimization, the performance of MoS with an equivalent parameter budget to LoRA with the rank of 8 remains competitive with that of LoRA with the rank of 64. This validates an 8x parameter reduction achieved by MoS again, aligning with our previous conclusions. Meanwhile, compared to LoRA, MoS exhibits comparable (even lower on average) standard deviations, and provides a higher average value, indicating **its similar stability and better performance.** These results will be further supplemented into our final paper!
> >
> > Avg. Performance:
> >
> >
> > | **Method** | **Rank** | **# Param.** | **MMLU**  | **BBH**  | **GSM8K**  | **TyDi QA**  | **TyDi QA** | **HumanEval** | **Avg.** |
> > |------------|----------|--------------|-----------------------|---------------------|-----------------------|----------------------------------|----------------------------------|----------------------------|----------|
> > | **LoRA**   | 8        | 12.16M       | 52.35                | 38.74              | 37.19                | 63.46                            | 46.09                            | 30.94                      | 44.79    |
> > |            | 64       | 97.26M       | 52.36                | 39.70              | 37.98                | 64.04                            | 46.53                            | 31.83                      | 45.41    |
> > | **MoS**    | 16       | 12.16M       | 52.41                | 40.14              | 37.89                | 63.78                            | 46.31                            | 31.78                      | 45.38    |
> >
> >
> > std:
> >
> >
> > | **Method** | **Rank** | **# Param.** | **MMLU**  | **BBH**  | **GSM8K**  | **TyDi QA**  | **TyDi QA** | **HumanEval** | **Avg.** |
> > |------------|----------|--------------|-----------------------|---------------------|-----------------------|----------------------------------|----------------------------------|----------------------------|----------|
> > | **LoRA**   | 8        | 12.16M       | 0.25                  | 0.89                | 1.59                  | 1.06                             | 1.14                             | 0.21                       | 0.86     |
> > |            | 64       | 97.26M       | 0.23                  | 0.52                | 0.72                  | 1.51                             | 1.89                             | 0.21                       | 0.85     |
> > | **MoS**    | 16       | 12.16M       | 0.17                  | 1.01                | 0.71                  | 1.05                             | 1.21                             | 0.23                       | 0.73     |

---

> > > ### Author Response · Authors · 2024-11-25
> > > **Response to Reviewer Gvvt (3/n)**
> > >
> > > **Question 2 & 3: Impact of random initialization & Robustness of hyper-parameters.**
> > >
> > > - As shown above, compared to LoRA, MoS exhibits comparable (even lower on average) standard deviations, and provides a higher average value, indicating its similar stability and better performance. In practice, **we also do not find any apparent impact of initialization on the performance of MoS.** Due to our limited computational resources, we do not extensively validate the performance of various hyperparameters to identify a consistent configuration. Instead, we optimize them separately for each task, given the diverse tasks for a comprehensive evaluation.
> > > - **To demonstrate the robustness of hyperparameters,** we perform a grid search for the configurations of private rank and shard number with 4 seeds (i.e., 0, 1, 2, 3). As detailed below, as the number of shards increases to provide more differentiation, the optimal private rank tends to decrease. **Generally, shard numbers of 4 or 8 yield the best results.** From another perspective, for any given private rank, there always exists a suitable range of shard numbers that consistently produce remarkable results (i.e., >=39.8%), **demonstrating the robustness of private rank.** We will also incorporate this analysis in our final paper!
> > >
> > >
> > > |           |    |     |    Private Rank   |       |       |
> > > |:---------:|:--:|:---------------------:|:-----:|:-----:|:-----:|
> > > |           |    |           1           |   3   |   5   |   7   |
> > > |           |  1 | 38.9%         | 39.3% | 39.1% | 39.3% |
> > > |           |  2 |         38.6%         | 39.3% | 39.7% | 39.5% |
> > > |   **Shards per Vector**        |  4 |         39.3%         | 39.8% | 40.0% | 39.6% |
> > > |           |  8 |         39.6%         | 39.8% | 39.6% | 38.8% |
> > > |           | 16 |         39.8%         | 39.3% | 39.3% | 38.6% |
> > >
> > >
> > > **Question 4: Benefit comparison of differentiation strategies.**
> > >
> > > - As discussion in the Sec. 4.4 of our submission, we conduct an ablation study on each differentiation strategy, and **compare their relative significance.** Briefly, all differentiation strategies contribute to improving the parameter efficiency of MoS, despite all being nearly cost-free. Among them, pair dissociation and shard privatization unleash the efficiency more saliently through increased combination diversity and exclusive differentiation, respectively, while vector sharding offers incremental gains. We recommend you refer to this section for further details.
> > >
> > > **Question 5: Computational overhead of MoS.**
> > >
> > > - **Training stage.** We finetune the LLaMA3.2-3B model for one epoch across various tasks, and record the time consumption (in hours) of LoRA and MoS in the following table. It is worth noticing that MoS does incur 2.80% more finetuning time. This is mainly caused by the increased rank for better performance. We do not find a free lunch here, and welcome any suggestive discussions!
> > > - **Finetuning stage.** At the design stage of MoS, we have **already tried to avoid any apparent drawbacks, and made it to be a nearly plug-and-play alternative to LoRA.** Hence, compared to LoRA, MoS only introduces a routing operation to form the low-rank matrices, whose computational overhead should be negligible. Besides, we intentionally adopt the index-based routing mechniasam so that **precomputing can be used** to prepare the low-rank matrices in parallel to the computation of preceding transformer blocks. This could **circumvent the latency of routing operation, and apply all the existing inference techniques of LoRA.** Due to the independence between models and the similarity with LoRA, MoS can seamlessly adapt to multi-model scenarios, and keeps suitable for the above analysis.
> > >
> > >
> > > | Method | Rank | Parameters | MMLU | BBH | GSM | Codex-Eval | Avg. |
> > > | ------ | ---- | ---------- | ---- | --- | --- | ---------- | ---  |
> > > | LoRA   | 8    | 12.16M     | 1.50 | 1.47 | 1.82 | 0.21       | 1.25   |
> > > | MoS    | 8    | 12.16M     | 1.54 | 1.52 | 1.86 | 0.22       | 1.285   |
> > >
> > >
> > > **Question 6: Trade-offs between parameter savings and performance across tasks.**
> > >
> > > - To the best of our knowledge, the initial motivation of MoS is to improve the parameter efficiency of LoRA, which means better performance with the same trainable parameter budget, or the same performance with fewer trainable parameters. Hence, **no trade-offs between parameter savings and performance have been observed with MoS.**
> > >
> > > Thank you again for your appreciation of our work and your detailed comments! They do help us to further polish our papers, and we also hope that our clarifications can also address your concerns! If you have any further questions, please feel free to let us know; we welcome any suggestions!
> > >
> > > Sincerely,
> > >
> > > Authors

---

> ### Comment · Reviewer_Gvvt · 2024-11-26
>
> Thank you for providing detailed responses to my comments. I have carefully reviewed all the rebuttal text. While I appreciate the additional context and clarifications, they do not change my assessment of the paper. Therefore, I intend to maintain my current score.

---

> > ### Author Response · Authors · 2024-11-27
> > **Response to Reviewer Gvvt**
> >
> > Dear Reviewer Gvvt,
> >
> > We sincerely appreciate **your thoughtful feedback, recognition of our work, and, especially, your great dedication to both our reviewing and rebuttal process!** Thank you!
> >
> >
> > Sincerely,
> >
> > Authors

---

### Official Review · Reviewer_opL8 · 2024-11-04

**Soundness:** 4
**Presentation:** 3
**Contribution:** 3
**Rating:** 6
**Confidence:** 5

**Summary:**

The paper introduces a novel fine-tuning method called **Mixture of Shards (MoS)**, which aims to significantly improve parameter efficiency in adapting large language models for customized applications. As large language models (LLMs) continue to scale, there is a growing need for parameter-efficient fine-tuning techniques to manage the high GPU memory overhead associated with serving multiple customized models simultaneously. Traditional approaches, such as Low-Rank Adaptation (LoRA), reduce resource consumption by updating pretrained weights with trainable low-rank matrices, but they still encounter scalability and memory limitations when applied to large models and extensive user customization. MoS offers a solution that retains the advantages of LoRA while achieving greater parameter efficiency through innovative parameter sharing and differentiation mechanisms.

The central concept behind MoS is to combine **inter-layer and intra-layer parameter sharing** in a single framework. This sharing is further enhanced by four lightweight differentiation strategies designed to counteract potential performance degradation from pure parameter sharing. These strategies include **subset selection**, **pair dissociation**, **vector sharding**, and **shard privatization**, each providing unique ways to increase the diversity and exclusivity of shared parameters across layers. By using a **Mixture-of-Experts (MoE)-like routing mechanism**, MoS selects and concatenates specific shards from a global parameter pool, thereby achieving efficient memory usage while maintaining high model performance.

In terms of experimental validation, the paper presents extensive evaluations on various NLP tasks, including factual knowledge (MMLU), multilingual question-answering (TyDi QA), mathematical reasoning (GSM8K), multi-step reasoning (BBH), and coding (HumanEval). The experiments demonstrate that MoS outperforms LoRA and other baseline methods in parameter efficiency, particularly under limited parameter budgets. MoS achieves approximately eightfold parameter savings compared to standard LoRA configurations, making it a promising approach for scenarios requiring numerous custom models.

An ablation study further examines the importance of each differentiation strategy, showing that components like pair dissociation and shard privatization provide substantial gains in efficiency, while vector sharding offers incremental improvements. The study reinforces the necessity of each differentiation strategy in achieving the performance and efficiency benefits observed with MoS. Additionally, a scalability analysis using the larger LLaMA2-13B model demonstrates that MoS maintains its advantages on a larger scale, further underscoring its robustness and suitability for high-capacity models.

The paper positions MoS as an important step forward in parameter-efficient fine-tuning. MoS’s compatibility with LoRA-based infrastructure and its ability to serve multiple customized models simultaneously without substantial memory overhead make it practical for real-world deployment. The findings provide insights into the trade-offs and design considerations of parameter sharing, offering a valuable resource for researchers and practitioners working on efficient model adaptation techniques. The paper’s detailed methodology, comprehensive experimentation, and focus on parameter efficiency contribute meaningfully to the broader research area of resource-efficient machine learning, addressing critical scalability issues as the field advances.

**Strengths:**

The paper provides solid technical grounding for the Mixture of Shards (MoS) method, with each component—inter-layer and intra-layer sharing and differentiation strategies. The Mixture of Shards (MoS) approach is a novel, well-motivated response to the growing need for efficient fine-tuning techniques for large models. By blending inter-layer and intra-layer sharing with lightweight differentiation strategies, the paper introduces a resource-efficient method that extends beyond existing parameter-sharing methods like LoRA, VeRA, and PRoLoRA. This innovative combination of techniques in MoS is practically a valuable approach.

The experimental design is comprehensive and addresses key aspects of parameter efficiency, memory usage, and model performance across a range of NLP benchmarks (e.g., MMLU, GSM8K, TyDi QA). The thorough ablation study underscores the necessity of each differentiation strategy (subset selection, pair dissociation, vector sharding, and shard privatization) and supports the paper’s claims about MoS’s efficiency. The paper also includes scalability tests, demonstrating MoS’s robustness on larger models, such as LLaMA2-13B, reinforcing its applicability to current large model architectures.

MoS integrates four nearly cost-free differentiation strategies—subset selection, pair dissociation, vector sharding, and shard privatization—to counteract the performance limitations of pure parameter sharing. These strategies is carefully designed to enhance the diversity and exclusivity of shared parameters, which contributes to the robustness and performance of the method.

The paper includes rigorous experimentation across diverse NLP benchmarks, such as MMLU ( Massive Multitask Language Understanding for factual knowledge), TyDi QA (multilingual question-answering), GSM8K (for mathematical reasoning), BBH (Big-Bench-Hard for multi-step reasoning), and HumanEval. These benchmarks test the model on factual knowledge, multilingual capabilities, mathematical reasoning, general reasoning, and coding. The results demonstrate MoS’s parameter efficiency and effectiveness compared to baseline methods, making a strong case for its practical utility. The parameter savings—approximately eightfold compared to standard LoRA—are significant, supporting the method’s scalability. This reduction substantially alleviates the memory burden, enabling more efficient model customization and serving without sacrificing performance, which is particularly valuable in settings requiring multiple concurrent models.

The paper is well-structured, with a logical flow that introduces the problem, presents the MoS solution, and discusses experimental results comprehensively. The clarity of the writing is generally good, though the differentiation strategies could benefit from additional diagrams or illustrations to aid in understanding for a wider audience.

**Weaknesses:**

While MoS is evaluated on a range of NLP tasks, the paper does not sufficiently analyze the method’s performance across various model architectures or specific task categories (e.g., multilingual tasks, code generation) where parameter efficiency and differentiation strategies could have different impacts. A breakdown showing how MoS performs on individual tasks, especially ones that are highly memory-intensive, would offer a clearer picture of its advantages and limitations across diverse NLP applications.

The ablation study is a strong point but could be enhanced by further exploring each differentiation strategy’s scaling potential. For instance, while the study confirms the individual benefits of subset selection, pair dissociation, vector sharding, and shard privatization, it doesn’t analyze the interactions or scalability of these strategies as model or task complexity increases. Additional experiments showing the performance impact of these strategies in larger configurations or different combinations would make the study more informative for readers looking to fine-tune MoS to specific needs.

The paper would benefit from a section discussing the potential limitations of MoS in specific scenarios. For instance, the effectiveness of MoS might be reduced when applied to tasks with low data diversity or high variance in representational needs across layers. Discussing scenarios where MoS might underperform or require adaptation would provide a more balanced view and help users assess when MoS is a suitable choice.

**Questions:**

Have you considered how MoS might interact with techniques like quantization, pruning, or dropout? Many practical deployments of large models use these methods in conjunction to manage resource constraints, and understanding how MoS might complement them would add value. If feasible, a brief experimental analysis or discussion on this integration would enhance the paper’s relevance for real-world applications.

Could you discuss potential limitations of MoS, such as scenarios where the method may underperform or require additional tuning? For example, does MoS have any specific limitations when applied to domains with high variability in representation needs across layers? A discussion on this would offer a more balanced perspective, helping readers assess the suitability of MoS in various contexts.

Has MoS been evaluated for its impact on inference latency, especially in multi-model serving scenarios?

Are there limitations to the size or complexity of models that MoS can handle effectively? For example, do the benefits of MoS start to diminish for models larger than LLaMA2-13B, or do you anticipate any challenges in scaling it to models with trillions of parameters?

---

> ### Author Response · Authors · 2024-11-25
> **Response to Reviewer opL8**
>
> Dear Reviewer opL8,
>
> We are grateful for your recognition of our efforts and the time you dedicated to reviewing our work! After carefully considering your feedback, we provide the following explanations with the hope of fully addressing your concerns!
>
> **Weekness 1: Insufficient analysis.**
>
> - For task categories, we **intentionally choose diverse tasks** with the hope of assessing the abilities of MoS as comprehensively as possible. As shown in the Table 2 of our submission, these tasks **have already covered your mentioned tasks**  (e.g., multilingual tasks, code generation), and MoS exhibits consistent improvement of parameter efficiency on both tasks. For various models, we choose LLaMA-2 7B and 13B models, and supplement the results of LLaMA-3.2 3B to **cover diverse model sizes.**  Besides, methods of LoRA series are quite general, and mainly impact the representation capacity for finetuning, **thereby being robust for tasks and models.**
> - For your reference, we further conduct the experiments with the LLaMA3.2-3B model. As shown in the following table, the performance of MoS with an equivalent parameter budget to LoRA with the rank of 8 remains competitive with that of LoRA with the rank of 64. This **validates an 8x parameter reduction achieved by MoS again, aligning with our previous conclusions.**
>
>
> | **Method** | **Rank** | **# Param.** | **MMLU**  | **BBH**  | **GSM8K**  | **TyDi QA**  | **TyDi QA** | **HumanEval** | **Avg.** |
> |---------|----------|--------------|---------|------|----|------|-------|----------|----------|
> | **LoRA**   | 8        | 12.16M       | 52.35                | 38.74              | 37.19                | 63.46                            | 46.09                            | 30.94                      | 44.79    |
> |            | 64       | 97.26M       | 52.36                | 39.70              | 37.98                | 64.04                            | 46.53                            | 31.83                      | 45.41    |
> | **MoS**    | 16       | 12.16M       | 52.41                | 40.14              | 37.89                | 63.78                            | 46.31                            | 31.78                      | 45.38    |
>
>
> **Weekness 2: Scalability on model or task complexity.**
>
> - As mentioned above, our analysis **covers different model sizes,** demonstrating the robusteness of MoS with regard to model scale. As for task complexity, we **conduct a comprehensive assessment with diverse datasets, covering different aspects of capacities.** The consistent enhancement validates the superiority of MoS across different tasks. We welcome any open discussion on this topic! If you have any suggestions, please feel free to let us know, and we can supplement relevant experiments.
>
> **Weekness 3: Potential limitations of MoS.**
>
> Honestly, we have **already tried to avoid any apparent drawbacks when designing MoS**, and find the following left overheads:
>
> - **Slightly more GPU consumption for finetuning.** Compared to LoRA with the same trainable parameter count, MoS increases the rank several times, leading to similar GPU consumption to LoRA with the same rank. However, MoS does provide better performance. We also finetune the LLaMA3.2-3B model for one epoch across various tasks, and record the time consumption (in hours) of LoRA and MoS in the following table. It is worth noticing that MoS only incur 2.80% more finetuning time. We tried to solve it, but haven't found a good solution for a free launch. If you're interested, we welcome any constructive discussions!
> - An inappropriate routing mechanism may incur more inference latency. Targeting this drawback, we give up existing activation-based routing mechanisms, because the routing operation has to wait for the activations, resulting in larger latency. Instead, we intentionally adopt the index-based routing mechanism so that precomputing can be used to prepare the whole low-rank matrices in parallel to the computation of preceding transformer blocks. This **could circumvent the latency of routing operation**, and apply all the existing inference techniques of LoRA.
> - For tasks with low data diversity or high representational variance, we actually cannot assert the conclusion without substantial experiments, due to the difficulty of controlling these elements for fair comparison. We also tried to find papers related to this analysis, but did not find suitable ones. Intuitively, **the equivalent rank can reduce the number of trainable parameters for easy tasks (e.g.,  low data diversity)**, while **the private rank and shard number per vector can adapt MoS for tasks requiring diverse representational power across layers.**
>
> | Method | Rank | Parameters | MMLU | BBH | GSM | Codex-Eval | Avg. |
> | ------ | ---- | ---------- | ---- | --- | --- | ---------- | ---  |
> | LoRA   | 8    | 12.16M     | 1.50 | 1.47 | 1.82 | 0.21       | 1.25   |
> | MoS    | 8    | 12.16M     | 1.54 | 1.52 | 1.86 | 0.22       | 1.285   |

---

> > ### Author Response · Authors · 2024-11-25
> > **Response to Reviewer opL8 (2/n)**
> >
> > **Question 1: Interaction with quantization, pruning, or dropout.**
> >
> > - More broadly, MoS focuses on improving the parameter efficiency of LoRA, and should be categorized into LoRA series. Since we maintain MoS to be a nearly plug-and-play alternative to LoRA, **MoS shares similar interactions with these techniques, as LoRA does.** Specifically, QLoRA [1] is one of the most famous combinations of LoRA and quantization methods, which employs LoRA, NF4 format, Double Quantization, and Paged Optimizers to restore the performance loss incurred by quantization for lightweight finetuning. In contrast, there are less combinations between pruning and LoRA. For example, APT [2] dynamically adds tuning parameters for fast and accurate convergence, while performing structured pruning for efficiency. A comprehensive study of how Dropout can contribute to LoRA-based Parameter Efficient Fine-Tuning is conducted by [3]. All these interactions can also be supported by MoS, since it is a nearly plug-and-play alternative to LoRA.
> >
> > **Question 2: Potential limitations of MoS.**
> >
> > - We elaborate the explanations in "Weekness 3: Potential limitations of MoS". Please refer to it for details.
> >
> > **Question 3: Inference latency.**
> >
> > - At the design stage of MoS, we have already tried to avoid any apparent drawbacks, including the inference latency. Targeting this point, we intentionally adopt the index-based routing mechanism so that precomputing can be used to prepare the whole low-rank matrices in parallel to the computation of preceding transformer blocks. **This could circumvent the latency of routing operation, and apply all the existing inference techniques of LoRA.** Due to the independence between models and the similarity with LoRA, MoS can seamlessly adapt to multi-model scenarios, and keeps suitable for the above analysis.
> >
> > **Question 4: Scalability.**
> >
> > We elaborate the explanations in "Weekness 1: Insufficient analysis". Please refer to it for details. Briefly, we **validate a consistent 8x parameter reduction achieved by MoS across diverse datasets and models.** We further conduct additional experiments on LLaMA3.2-3B, where MoS exhibits the same parameter saving. Due to the limitations on computational resources, we haven't conducted experiments on larger models. As discussed in the Introduction section of our submission, with the scaling of models, the parameter efficiency of LoRA needs to be further enhanced, which **necessitates higher parameter efficiency of MoS.** Given the generalizability of LoRA-like methods, MoS should also be scalable to larger models.
> >
> > Thank you again for your recognition of our work! If you have any further questions, please feel free to let us know; we welcome any insightful discussions!
> >
> > Sincerely,
> >
> > Authors
> >
> > **References**
> >
> > [1] Tim Dettmers, Artidoro Pagnoni, Ari Holtzman, and Luke Zettlemoyer. Qlora: Efficient finetuning of quantized llms. arXiv preprint arXiv:2305.14314, 2023.
> >
> > [2] Bowen Zhao, Hannaneh Hajishirzi, and Qingqing Cao. Apt: Adaptive pruning and tuning pretrained language models for efficient training and inference. arXiv preprint arXiv:2401.12200, 2024.
> >
> > [3] Wang, S., Chen, L., Jiang, J., Xue, B., Kong, L., and Wu, C., "LoRA Meets Dropout under a Unified Framework", arXiv preprint arXiv:2403.00812.

---

### Official Review · Reviewer_m4F3 · 2024-11-05

**Soundness:** 4
**Presentation:** 3
**Contribution:** 3
**Rating:** 8
**Confidence:** 3

**Summary:**

The paper investigates a more lightweight solution than LoRA in order to serve a large number of finetuned models at the same time. Based on a finding that excessive sharing may hinder model performance, the authors believe that differentiation is necessary to reverse the detrimental effects of pure sharing. The paper proposes Mixture of Shards (MoS) that incorporates both inter-layer and intra-layer sharing with four differentiation strategies.

- Subset selection: randomly choose r pairs of vectors at the beginning for each layer
- Pair dissociation: separate each pair into two pools, where each vector in a pair will be sampled independently
- Vector sharding: shard the global pool into n parts and concatenate sampled vectors from each shard
- Shard privatization: reserve a private segment for exclusive use for each matrix

**Strengths:**

- The method is more general than LoRA, making LoRA a special case when there is no global pool.
- The authors provide ablation study for each of the differentiation strategies (except subset selection), showing the efficacy of each strategy.
- Overall, I find the finding about sharing & differentiation makes sense and the motivation is clear. Each differentiation strategy is proposed to keep the number of parameters unchanged but increase level of differentiation between each layer.

**Weaknesses:**

Overall, the paper is well written. There are some minor details that can be improved.
- Figure 2 can be more accurate and following the main text better. There is no mentioning of router "R" in the main text. Notations like $A^{pub}$, $A^{pri}$, $B$, $I$, $m_{ij}$ can be used to make it clearer.
- Index(.) could be replaced by a better notation since .index(.) can be understood as the index of some element in an array.

**Questions:**

- In Section 3.3, is $I_a^k \in \mathbb{R}^r$ or $\mathbb{N}^r$
- What do the 4/8, 16/32 (or "increasing the rank to 4 or 8") in Table 2 mean?
- Many implementation details are missing - what's the pool size, how many shards, breakdown of private & public segments, etc.

---

> ### Author Response · Authors · 2024-11-25
> **Response to Reviewer m4F3**
>
> Dear Reviewer m4F3,
>
> We sincerely thank you for your great appreciation on our work and the time you spent on it! We have read your comments carefully and made the following clarifications. Hope that they could further address your concerns!
>
> **Weakness 1 & 2: Refined notations.**
>
> Thank you so much for pointing out the confusing flaws in our papers! We will follow your suggestions to refine them one by one, including consistent fonts, extra notations, explanation of router "R" in the main text, and usage of .index(.).
>
> **Question 1: Refined notation.**
>
> Yes, you are right, and Thanks! Both $I^k_a
> $ and $I^k_b$ should be in $\mathbb{N}^r$. We will correct this!
>
> **Question 2: Explanation of "4/8" and "16/32".**
>
> Here we follow the notations in PRoLoRA, an important baseline for our method. Specifically, for fair comparison, we keep an identical trainable parameter budget for both LoRA, PRoLoRA and MoS, treat the raised rank of PRoLoRA and MoS as a hyper-parameter, and report the results in one line for more intuitive comparison. "4/8" and "16/32" indicate raising the rank to either 4 or 8, and 16 or 32, respectively. Thank you for raising this question! We will also supplement further clarification, and employ an extra symbol to differentiate them in our final version.
>
> **Question 3: Missing implementation details.**
>
> Thanks for your kind reminder! Due to the page limit, we arrange the "Experiment Details" in the Appendix of our submission, and will double-check it to supplement further details. Besides, we attached our code in the supplementary materials. If needed, you can check it for details. We will also open-source our code and environment for better reproducibility with minimal efforts upon publication.
>
> Thank you again for your appreciation on our work! Also, your detailed comments help us further polish our paper. If you have any further questions, please feel free to let us know! Thank you!
>
> Sincerely,
>
> Authors

---

### Meta-Review · Area_Chair_8uCg · 2024-12-15

**Metareview:**

The paper presents a novel method called Mixture of Shards that combines inter-layer and intra-layer sharing with lightweight differentiation strategies to enhance parameter efficiency and model performance for large models. The MoS method outperforms existing approaches like LoRA by incorporating four differentiation strategies—subset selection, pair dissociation, vector sharding, and shard privatization—offering a resource-efficient solution that scales well with large models. The comprehensive ablation study and experiments across multiple NLP benchmarks demonstrate its robustness and significant parameter savings.

All reviewers acknowledge the contributions of the paper, with feedback generally leaning towards acceptance. The AC carefully reviewed the paper and rebuttal, agreeing with the recommendation for acceptance based on the consistency.

**Additional Comments On Reviewer Discussion:**

The reviewers noted that most of the concerns raised have been addressed, leading to a unanimous recommendation for acceptance.

---

### Decision · Program_Chairs · 2025-01-22

Accept (Poster)